# Do Multiple Instance Learning Models Transfer?

Daniel Shao [1 2]   Richard J. Chen [2]   Andrew H. Song [2]   Joel Runevic [2]   Ming Y. Lu [1 2]   Tong Ding [2]
Faisal Mahmood [2]

## Abstract

Multiple Instance Learning (MIL) is a cornerstone approach in computational pathology (CPath) for generating clinically meaningful slide-level embeddings from gigapixel tissue images. However, MIL often struggles with small, weakly supervised clinical datasets. In contrast to fields such as NLP and conventional computer vision, where transfer learning is widely used to address data scarcity, the transferability of MIL models remains poorly understood. In this study, we systematically evaluate the transfer learning capabilities of pretrained MIL models by assessing 11 models across 21 pretraining tasks for morphological and molecular subtype prediction. Our results show that pretrained MIL models, even when trained on different organs than the target task, consistently outperform models trained from scratch. Moreover, pretraining on pancancer datasets enables strong generalization across organs and tasks, outperforming slide foundation models while using substantially less pretraining data. These findings highlight the robust adaptability of MIL models and demonstrate the benefits of leveraging transfer learning to boost performance in CPath. Lastly, we provide a resource which standardizes the implementation of MIL models and provide model weights for FEATHER, a PC-108 pretrained ABMIL model at https://github.com/mahmoodlab/MIL-Lab.

## 1. Introduction

Multiple Instance Learning (MIL) has been the foundational paradigm in computational pathology (CPath) for over a decade. Digitized human tissue sections, referred to as

[1]Massachusetts Institute of Technology, Cambridge MA, USA [2]Harvard University, Boston MA, USA. Correspondence to: Daniel Shao <dshao@mit.edu>, Faisal Mahmood <FaisalMahmood@bwh.harvard.edu>.

*Proceedings of the $42^{nd}$ International Conference on Machine Learning*, Vancouver, Canada. PMLR 267, 2025. Copyright 2025 by the author(s).

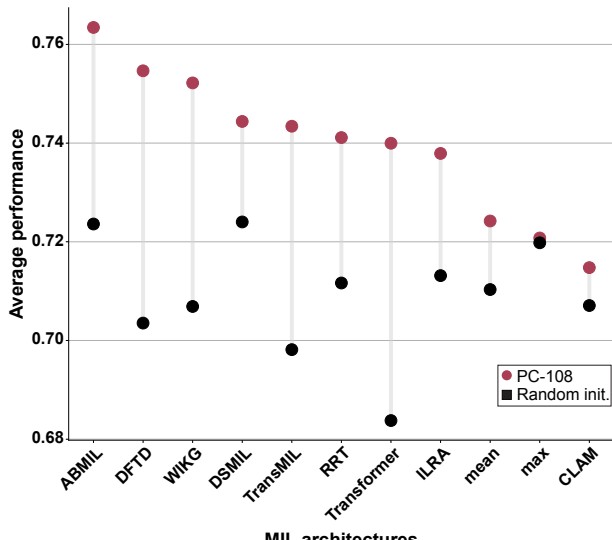

*Figure 1.* **Average performance with supervised pretraining vs. random initialization.** Performance of MIL models trained from random initialization (black) vs. initialized with weights from a model pretrained (red) on a 108-class pancancer task (n=3,944 WSIs). Performance is averaged across the 19 classification tasks, using AUROC for the binary classification tasks, weighted kappa for grading, and balanced accuracy for the multiclass tasks.

whole slide images (WSIs), pose unique challenges due to their gigapixel resolutions and sparse diagnostic regions. The weakly-supervised MIL framework addresses these challenges by (1) using a pretrained encoder to tokenize a WSI into patch-level features, and (2) using a trainable aggregator to pool these patch features into a final slide-level representation for downstream classification (Song et al., 2023; Campanella et al., 2024).

While substantial progress has been made in developing powerful pretrained patch encoders for pathology (Chen et al., 2024a; Vorontsov et al., 2024; Xu et al., 2024b; Lu et al., 2024), learning an effective aggregation scheme remains an open challenge due to the relatively small training cohorts and sparse diagnostic regions characteristic of CPath datasets. As a result, new MIL architectures have been continually emerging since 2015, with each approach introducing new inductive biases to improve data efficiency

and model generalizability during training.

Despite the overwhelming research interest in MIL architecture development and well-recognized benefits of transfer learning in low-data regimes in the biomedical community, surprisingly little is known about how well MIL models transfer in CPath. To date, random weight initialization remains a common standard in MIL model development and evaluation. Though random weight initialization is appropriate for ablating architecture hyper-parameters and developing academic benchmarks, given the enormous challenge of bringing pathology AI models to clinical adoption using small patient cohorts and widespread use of patch-level transfer in MIL, it is surprising that there is little empirical understanding of how well slide-level transfer performs, *e.g.* using a MIL model pretrained on one task and transferring it to another task with frozen feature evaluation or lightweight finetuning.

This knowledge gap is in stark contrast to the enormous progress made in general machine learning on answering many fundamental questions, such as: How does model architecture (Zoph et al., 2018; Kornblith et al., 2019; Zhou et al., 2021; Renggli et al., 2022), data scale (Kolesnikov et al., 2020), and pretext task (Tripuraneni et al., 2020; Ericsson et al., 2021; Zhao et al., 2021; Mehra et al., 2024) affect transfer performance? Does better supervised performance lead to better transfer? (Recht et al., 2019; He et al., 2019; Kumar et al., 2022; Fang et al., 2024) When does transfer not work? (Raghu et al., 2019; Jang et al., 2019; Pruksachatkun et al., 2020; You et al., 2021) And what features are being transferred? (Neyshabur et al., 2020)

Indeed, the transfer of MIL models is also pertinent in light of recent interest in whole slide foundation models, which aim to extract general-purpose slide-level representations that can transfer and generalize to challenging clinical tasks in low-data regimes (Shaikovski et al., 2024; Wang et al., 2024; Ding et al., 2024; Vaidya et al., 2025). Though self-supervised learning has demonstrated favorable properties for developing slide foundation models via data scaling, model generalization, and emergent capabilities, we contend that an MIL model trained with supervision on large-scale, diverse hierarchical classification tasks can also function as a slide foundation model. We hypothesize this approach can embed WSIs and generalize to diverse, challenging tasks, while requiring substantially less pretraining data than its self-supervised counterparts.

In this work, we investigate transfer learning capabilities of pretrained MIL models in CPath, evaluating 11 MIL models and using 21 tasks for supervised pretraining and transfer evaluation which spans many- and few-shot morphological classification, cancer grading, and biomarker prediction. This work provides a roadmap for how to transfer MIL models, answering: which MIL architectures transfer better,

whether models can transfer to different organs, disease indications, or task types, does transfer learning always outperform training from scratch, and more. We summarize our main findings below:

- Pretrained MIL models consistently outperform MIL models trained with randomly initialized weights, even when pretrained on out-of-domain tasks.
- Models pretrained on pancancer tasks are data-efficient and generalize effectively across organs and task types.
- Transfer performance varies with MIL architecture and model size. Larger models benefit more from pretraining, and pancancer pretraining unlocks favorable scaling trends.
- The aggregation scheme learned during pretraining is pivotal for observed gains from model transfer.

Finally, we share a GitHub library to standardize MIL implementation and simplify loading pretrained weights across various CPath tasks.

## 2. Related work

### 2.1. MIL in CPath

MIL is a form of supervised learning where models are trained on labeled collections of instances, known as "bags", without individual instance labels. In CPath, MIL has emerged as the predominant framework for modeling WSIs, where tissue regions from a WSI (bag) are first segmented and then divided into non-overlapping image patches (instances), with variable bag sizes of around 1,000 to 10,000 (Campanella et al., 2019; Lu et al., 2021). A MIL model learns a mapping from the collection of image patches of each slide to the slide-level labels, without supervision at the patch level. Numerous variations of MIL methods have been proposed, distinguishing from one another by the aggregator function (e.g. non-parametric max-pooling (Campanella et al., 2019) *vs.* learned parametric pooling (Ilse et al., 2018)), assumption in permutation invariance (e.g. modeling instances as an unordered set (Ilse et al., 2018) *vs.* an ordered sequence (Shao et al., 2021)), the precise model architecture (e.g. feedforward layers only (Lu et al., 2021; Ilse et al., 2018; Li et al., 2021) *vs.* transformer attention (Shao et al., 2021)) and the use of auxilliary objective functions during training (Lu et al., 2021; Zhang et al., 2022). We cover diverse types of architectures to derive general insights on MIL transfer.

### 2.2. Large-scale pretrained slide foundation models

With the increasing availability of WSIs, recent works have also explored pretraining MIL architectures to develop slide foundation models that produce general-purpose slide representations (Chen et al., 2022; Jaume et al., 2024; Wang et al., 2024; Xu et al., 2024a; Ding et al., 2024; Vaidya et al.,

2025). Instead of leveraging carefully curated labels for task-specific supervised MIL training, these approaches often rely on large-scale unimodal and multimodal data and task-agnostic objectives such as self-distillation and contrastive learning to learn representations that can be transferred to diverse downstream tasks. We provide a rigorous comparison of supervised pretraining against such slide foundation models.

# 3. Preliminaries

The primary goal of this work is to understand how well supervised MIL models transfer to various downstream tasks in computational pathology (CPath). This investigation is motivated by the overwhelming interest in developing slide foundation models, *e.g.* self-supervised MIL models that extract general-purpose slide representations with little-to-zero finetuning. In this context, supervised MIL with model weight transfer is a simple and fundamental alternative overlooked in the research community, and we hypothesize that transferring supervised MIL models may outperform current techniques and learning paradigms proposed for solving challenging CPath tasks. To this end, we exhaustively evaluate transfer performance of 11 MIL architectues across 19 publicly available benchmarks. We outline the evaluation protocol, pretraining and target datasets, and MIL architectures used for assessing MIL transfer below.

## 3.1. Supervised MIL Transfer

The experimental setup for evaluating supervised MIL transfer follows that of previous works in transfer learning, in which a network $f$ is first pretrained using a supervised task from source domain $\mathcal{D}_S$ (*pretrain task*) and then evaluated on a different task from target domain $\mathcal{D}_T$ (*target task*). We study the following 11 architectures for assessing MIL transfer: ABMIL (Ilse et al., 2018), CLAM (Lu et al., 2021), DSMIL (Li et al., 2021), DFTD (Zhang et al., 2022), TransMIL (Shao et al., 2021), Transformer (Wagner et al., 2023; Vaswani et al., 2023), ILRA (Xiang & Zhang, 2023), RRT (Tang et al., 2024), WIKG (Li et al., 2024), Mean-MIL and MaxMIL. We measure transfer performance to the target task using two evaluation settings: (1) end-to-end finetuning, and (2) frozen feature evaluation via K-nearest neighbors (KNN) on pre-extracted slide-level embeddings.

For evaluation, we assess MIL transfer performance on 19 publicly available CPath tasks, with training datasets ranging in size from 314 to 8,492 WSIs and in label complexity from 2 to 30 classes. To ensure that the observed trends in this study are not confined to a specific organ, disease, or type of task, we comprehensively evaluate on four organs (breast, lung, prostate, and brain), and diverse task types such as cancer classification, cancer grading, and molecular subtyping, which are used as both pretrain and target

tasks. For example, to assess the tranferrability of ABMIL pretrained on NSCLC subtyping, we would first train an ABMIL model from scratch on the NSCLC subtyping task, followed by end-to-end finetuning on the other 18 target tasks such as BRACS coarse-grained subtyping. For MIL architecture $f$, pretrain task $s$ and target task $t$, we perform the following main comparisons which answer:

- $f_{s \to t}$ vs. $f_{\text{random init} \to t}$: For the same MIL architecture $f$, does pretraining task $s$ outperform training from scratch on task $t$?
- $f_{s \to t}$ vs. $f_{s' \to t}$: For the same MIL architecture $f$, do different pretrain tasks $s$ and $s'$ transfer better to $t$?
- $f_{s \to t}$ vs. $f'_{s \to t}$: Using the same pretrain task $s$, do different MIL architectures $f$ and $f'$ transfer better to task $t$?

Though this experimental setup follows previous investigations in computer vision, one limitation in assessing model transfer in CPath is the lack of large-scale, diverse classication datasets similar to ImageNet-1k (IN-1k), which has been the standard pretraining task for assessing transfer of various image recognition backbones over the past decade. In addition to the 19 tasks, we also include two pan-cancer tasks called PC-43 and PC-108 (Chen et al., 2024a). These represent 43-class and 108-class cancer subtyping tasks encompassing diverse malignancies from 17 organ types and are curated from the same hierarchical classification dataset for pretraining purposes only. PC-43 and PC-108 consist of the same set of 3,499 WSIs with either coarse labels (43 main cancer types) or fine-grained labels (108 OncoTree codes), respectively, with at least 5 and 15 WSIs per OncoTree code for training and testing. Overall, these tasks were developed to emulate IN-1k in label complexity and flexibility for supervised pretraining, which we hypothesize to be similar to IN-1k in developing transferrable models.

## 3.2. Implementation Details

For WSI preprocessing, we use the following standardized protocol for comparisons across MIL models: $256 \times 256$ non-overlapping tissue patching at $20\times$ magnification (0.5 $\mu$m/pixel) followed by pre-extracting features using UNI, a DINOv2-pretrained ViT-L/16 encoder (Oquab et al., 2024; Chen et al., 2024a). Unless specified otherwise, all MIL models are implemented using the author's original model definition, trained with UNI features, and with standardized hyperparameters: AdamW optimzier with a learning rate of $1 \times 10^{-4}$, cosine decay scheduler, and a maximum of 20 epochs with early stopping patience of 5 epochs on the validation set. Further details are provided in **Section B.1**.

# 4. Results

## 4.1. Which pretrain tasks are best for MIL transfer?

We first characterize the pretraining task quality by assessing how well a model fully-trained on a task can generalize to new tasks with frozen weights. Though not intuitive from a clinical perspective how a model trained on brain tissue would lead to better performance on lung tissue, we hypothesize that tasks with diverse diagnostic entities may confer unique advantages by allowing the model to learn from a more diverse set of slide representations across pretraining and training. Additionally, many histological entities are consistently of low diagnostic relevance, such as smooth muscle, red blood cells, and tissue processing artifacts, just as the recognition of nuclei and immune cells remain pertinent across an array of tasks in CPath. These commonalities across tasks suggest that MIL models may learn highly transferable aggregation methods. We evaluate the quality

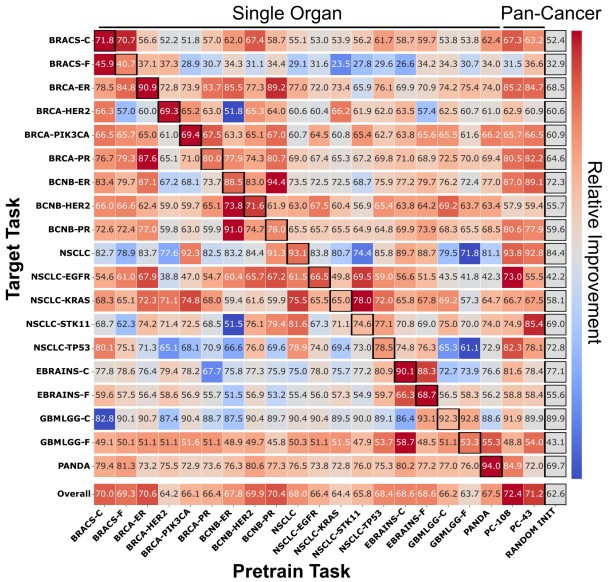

*Figure 2.* **Transfer performance across pretrain tasks**. The contingency table shows the average KNN performance of three MIL models (ABMIL, TransMIL, Transformer) transferring from the 21 pretrain tasks (columns, bottom axis) to the 19 target tasks (rows, left axis). The rightmost column (colored gray) shows the baseline of random weights. We use AUROC for binary classification, Cohen's weighted kappa for prostate grading, and balanced accuracy for multiclass classification tasks. The heatmap indicates the relative performance compared to baseline, with red indicating improvement and blue indicating performance decrease for each task.

of "off-the-shelf" slide representations from pretrained MIL models. To do so, we use a KNN classifier (k=20) to measure transfer performance from 21 pretraining tasks to 19 target tasks, averaging results across three MIL architectures

(ABMIL, TransMIL, Transformer). As shown in the contingency matrix in **Figure 2**, representations from pretrained models consistently outperform a random-weight baseline (improvement indicated by red in the matrix). We observe that pretraining on both in-domain and out-of-domain data yields strong results; for example, models pretrained on lung cancer (NSCLC) transferred to breast biopsy tasks (BCNB) as effectively as models pretrained on breast cancer (BRCA, BRACS). Notably, pancancer pretraining delivered the most substantial performance gains. Pretraining on PC-108 and PC-43 improves performance over the baseline an average of $+9.8\%$ and $8.6\%$, respectively. These findings suggest that pancancer pretraining is an effective strategy for obtaining highly transferable slide-level representations.

## 4.2. How does MIL architecture relate to transfer performance?

Using PC-108 as the representative pretraining task, we next perform a more comprehensive assessment of how well MIL architectures transfer. New MIL architectures are constantly emerging, each proposing new ways to model inductive biases such as cell-cell and cell-tissue relationships in the tissue microenvironment, yet it remains unclear whether these architectural innovations translate to improved generalizability or transferability. To gain further insights into this question, we compare the performance of 11 MIL architectures pretrained on PC-108 against their counterparts initialized with random weights. Aside from the initialization weights, all other training settings are fixed (**Section B.2**).

**Figure 1** and **Table 1** illustrate the average performance across all 19 tasks and performance grouped by distinct datasets, respectively. We observe that all MIL approaches benefit from pretraining, with an average improvement of 3.3% when using a pretrained PC-108 model compared to training from scratch. ABMIL is among the top-performing models across both initialization strategies, with a 3.8% average increase in performance. This aligns with recent findings that, given strong patch foundation models, simpler MIL approaches tend to outperform more complex alternatives (Chen et al., 2024a;b; Song et al., 2024). Additionally, DSMIL, a closely related method to ABMIL which adds patch-level predictions to ABMIL's global pooling method, attains the highest performance with random initialization. This further substantiates the efficacy of simpler weighted pooling approaches under random initialization.

While pancancer pretraining improves the performance of all models, the extent of improvement is architecture-dependent. For instance, we find that Transformer-based models, (TransMIL and Transformer), exhibit substantial improvements from pretraining, showing average improvements of 5.82%, and 5.83% respectively across all tasks. We hypothesize that because these architectures are highly

*Table 1.* **Finetuning performance across different MIL and datasets.** Performance with random initialization (Base) and PC-108 pretraining for each MIL framework, averaged within each task group. Reported metrics are balanced accuracy for multi-class tasks ($C > 2$) and weighted $\kappa$ for PANDA; AUROC otherwise. $\Delta$ denotes the performance difference between PC-108 and Base. Parentheses indicate the average standard deviation across the grouped tasks, with standard deviation determined by 1,000 bootstrap trials.

| Task | Init. | ABMIL | CLAM | DFTD | DSMIL | ILRA | RRT | TransMIL | Transformer | WIKG | maxMIL | meanMIL | Average |
|---|---|---|---|---|---|---|---|---|---|---|---|---|---|
| BRACS (2 tasks) | Base | 53.6(4.5) | 43.6(5.2) | 50.0(4.7) | 53.5(4.5) | 55.7(3.9) | 53.7(3.8) | 50.2(3.8) | 36.8(3.3) | 50.3(4.7) | 49.4(3.8) | 49.4(4.5) | 49.7 |
| | PC-108 | 57.8(4.9) | 44.6(4.8) | 57.9(4.7) | 57.0(4.5) | 51.8(3.9) | 52.4(4.2) | 50.7(5.0) | 53.2(5.3) | 62.1(4.5) | 48.9(4.1) | 41.4(4.9) | 52.5 |
| | $\Delta$ | +4.2 | +1.0 | +7.9 | +3.5 | −3.9 | −1.3 | +0.5 | +16.4 | +11.8 | -0.5 | −8.0 | +2.8 |
| BCNB (3 tasks) | Base | 80.9(4.5) | 80.5(4.8) | 78.6(5.1) | 79.6(4.5) | 79.0(4.4) | 78.3(4.6) | 76.9(4.6) | 75.8(4.9) | 83.4(4.1) | 83.0(4.2) | 79.8(4.0) | 79.6 |
| | PC-108 | 84.5(4.3) | 82.4(4.6) | 86.1(5.0) | 83.0(4.5) | 81.4(4.2) | 79.1(4.7) | 84.1(5.3) | 79.1(4.9) | 82.4(4.3) | 83.1(4.1) | 83.4(4.7) | 82.6 |
| | $\Delta$ | +3.6 | +1.9 | +7.5 | +3.4 | +2.4 | +0.8 | +7.2 | +3.4 | −1.0 | +0.1 | +3.6 | +3.0 |
| BRCA (4 tasks) | Base | 69.2(4.1) | 67.8(4.2) | 69.4(4.1) | 68.9(4.1) | 68.7(3.5) | 66.7(4.0) | 64.7(3.9) | 64.2(4.2) | 66.0(4.4) | 66.7(4.5) | 71.2(4.0) | 67.6 |
| | PC-108 | 73.3(3.6) | 65.1(4.1) | 75.5(4.2) | 69.4(3.5) | 72.2(3.6) | 70.4(4.0) | 72.1(3.8) | 71.4(3.8) | 69.1(4.5) | 70.9(3.8) | 70.7(4.2) | 70.9 |
| | $\Delta$ | +4.1 | −2.7 | +6.1 | +0.5 | +3.5 | +3.7 | +7.4 | +7.2 | +3.1 | +4.2 | -0.5 | +3.3 |
| EBRAINS (2 tasks) | Base | 76.6(2.2) | 78.8(2.1) | 76.6(1.9) | 75.2(1.9) | 78.2(2.0) | 77.6(1.9) | 72.2(2.2) | 76.2(2.0) | 75.1(2.0) | 73.4(2.2) | 78.8(2.5) | 76.2 |
| | PC-108 | 78.4(2.1) | 80.1(2.1) | 80.3(1.9) | 77.2(1.9) | 76.8(2.0) | 78.4(2.0) | 79.6(2.1) | 79.2(2.0) | 78.2(2.0) | 79.6(2.1) | 80.1(2.0) | 78.9 |
| | $\Delta$ | +1.8 | +1.3 | +3.7 | +2.0 | −1.4 | +0.8 | +7.4 | +3.0 | +3.1 | +6.2 | +1.3 | +2.6 |
| GBMLGG (2 tasks) | Base | 68.9(2.5) | 66.6(2.8) | 67.8(2.6) | 72.2(2.3) | 73.6(2.4) | 71.0(2.6) | 69.7(2.9) | 68.9(2.4) | 67.0(2.4) | 72.2(2.5) | 70.5(2.3) | 69.9 |
| | PC-108 | 73.6(2.1) | 71.7(2.2) | 74.4(2.1) | 75.4(2.1) | 70.2(2.1) | 74.1(2.2) | 71.9(2.3) | 70.2(2.3) | 67.9(2.1) | 73.0(2.1) | 70.4(2.2) | 72.1 |
| | $\Delta$ | +4.7 | +4.1 | +6.7 | +3.2 | −3.4 | +3.1 | +2.2 | +1.3 | +0.9 | +0.8 | −0.1 | +2.2 |
| NSCLC Morph (1 task) | Base | 95.3(0.6) | 91.1(0.8) | 92.1(0.7) | 94.0(0.9) | 90.4(0.7) | 93.9(0.7) | 91.3(0.8) | 93.3(0.8) | 91.3(0.8) | 95.9(0.5) | 91.1(0.7) | 92.7 |
| | PC-108 | 96.1(0.6) | 92.0(0.8) | 96.6(0.7) | 95.3(0.9) | 94.8(0.7) | 94.7(0.7) | 95.4(0.7) | 94.4(0.7) | 94.7(0.7) | 95.4(0.6) | 92.3(0.7) | 94.7 |
| | $\Delta$ | +0.8 | +0.9 | +4.5 | +1.3 | +4.4 | +0.8 | +4.1 | +1.1 | +3.4 | −0.5 | +1.2 | +2.0 |
| NSCLC Molec (4 tasks) | Base | 67.4(5.2) | 67.5(6.4) | 64.0(6.9) | 68.3(7.3) | 62.8(6.5) | 67.2(7.2) | 66.1(6.6) | 66.1(6.8) | 62.4(6.6) | 69.1(7.3) | 67.4(6.3) | 66.2 |
| | PC-108 | 72.8(5.2) | 68.5(6.3) | 74.3(7.2) | 68.8(6.3) | 71.0(6.1) | 73.3(6.4) | 75.0(6.2) | 73.2(4.9) | 75.0(6.1) | 62.4(6.6) | 74.1(6.3) | 71.7 |
| | $\Delta$ | +5.4 | +1.0 | +10.3 | +0.5 | +8.2 | +6.1 | +8.9 | +7.1 | +12.6 | −6.7 | +6.7 | +5.5 |
| PANDA (1 task) | Base | 91.6(0.7) | 91.2(1.0) | 87.9(0.7) | 91.2(0.7) | 91.5(0.9) | 91.1(0.9) | 90.5(0.9) | 90.4(1.0) | 92.1(0.8) | 89.7(0.9) | 91.2(1.0) | 90.8 |
| | PC-108 | 93.3(0.5) | 91.8(1.0) | 93.2(0.7) | 93.5(0.7) | 91.5(0.9) | 91.5(0.9) | 89.9(0.8) | 90.7(0.8) | 93.0(0.8) | 88.9(0.9) | 90.9(1.0) | 91.8 |
| | $\Delta$ | +1.7 | +0.6 | +5.3 | +2.3 | 0.0 | +0.4 | −0.6 | +0.3 | +0.9 | −0.8 | −0.3 | +1.0 |

parameterized, they are also more liable to overfit the training set, and may consequently benefit more from effective initializations. We directly investigate this relationship between model size and transfer performance in **Section 4.4**.

Meanwhile, the non-parametric aggregation models (mean-MIL and maxMIL) exhibit comparatively small changes in performance, suggesting that the transfer of knowledge between aggregation components is the key ingredient of MIL transfer. We also found that DSMIL and CLAM, closely related due to reliance on auxiliary loss for classifying key instances, exhibit lower improvements from transfer compared to other methods. We hypothesize that since the instance-level auxiliary loss guides the training dynamics of CLAM and DSMIL on top of the original slide-level loss, the initial weights are less pertinent to the model convergence, thereby reducing the benefit of MIL transfer.

Furthermore, we observe that the best performance from random initialization (72.3 with DSMIL) is lower than 9 of the 11 pretrained models, suggesting that the use of an effective initialization plays a more important role in performance than the quality of the MIL method. While many techniques have been proposed for developing new MIL architectures, we find that most techniques do not demonstrate any significant margin of improvement over ABMIL. Altogether, these results indicate that, rather than further architectural innovations, high-quality pretraining of simple but effective MIL models leads to the most performant models. To confirm the robustness of these results, we repeated experiments across five random seeds for ABMIL, DFTD, TransMIL, and RRT (**Table A4**). In all cases, the performance gains from pretraining remained consistent, reinforcing the reliability of these gains from pancancer pretraining.

### 4.3. Pretrained models are few-shot learners

An important consideration for MIL methods is learning with only a limited number of samples, a common scenario when dealing with rare diseases (Huang et al., 2023; Lu et al., 2024; Chen et al., 2024a). We investigate the data efficiency of transferred MIL models by probing performance in few-shot scenarios. Specifically, we randomly sample $K \in \{4, 16, 32\}$ samples from each class of $C$ diagnostic classes to construct a dataset of $K \cdot C$ samples, and train five different MIL methods (ABMIL, Transformer, TransMIL, DFTD, CLAM) on the molecular subtyping tasks of NSCLC-TP53/STK11/EGFR, BCNB ER/PR/HER2, and GBMLGG-C. The experiments are repeated five times to mitigate sampling bias. To delineate the effects of pretraining datasets, we ablate over random initialization, PC-43, and PC-108 pretraining datasets.

Across all five methods in **Figure 3**, we observe a clear trend of pancancer pretraining outperforming random initialization across all shots. The difference gap is especially pronounced for a lower number of shots for all MIL models, with DFTD boasting a 171% increase for $K = 4$ over random initialization. This underscores the data-efficiency and potential for MIL transfer to assist in data-sparse regimes. Between PC-108 and PC-43, we observe that PC-108 obtains higher performance at every configuration, suggesting that the models trained with a more challenging fine-grained classification task exhibit better data efficiency.

### 4.4. Do larger MIL architectures transfer better?

As a subsequent step to understanding transfer trends *across* different MIL models, we investigate the transfer trends

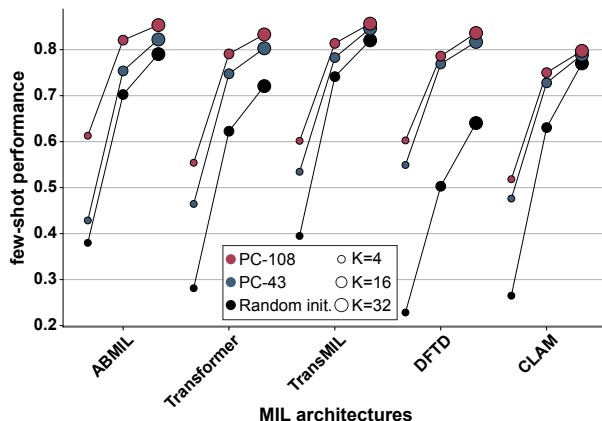

Figure 3. **Few-shot performance**. Few-shot performance with different initialization strategies for $K = \{4, 16, 32\}$ samples over five MIL methods on molecular subtyping tasks. Performance is averaged over test splits of 5-fold cross-validation.

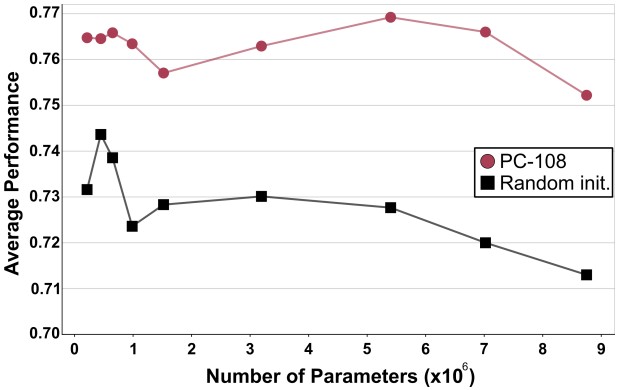

Figure 4. **Transfer at different model scales**. Average performance of different ABMIL scales across 19 evaluation tasks with initialization from random weights and PC-108 pretraining.

*within* a MIL model, by analyzing the relationship between transfer performance and model scale. With ABMIL demonstrating the best MIL transfer performance, we down- or up-scale the model size of ABMIL ($\approx 9 \times 10^5$ parameters), by adjusting the depth and width of the pre-attention MLP, with changes described in **Table A2**.

We observe that PC-108 pretraining consistently and substantially outperforms random initialization at all model scales (**Figure 4**). Furthermore, while random initialization led to variable performance across different model scales, the pretrained model performance remains comparatively stable across different model scales, with a monotonic increase from 0.1 million to 5 million parameters, and subsequently decreasing in performance at 9 million parameters. This consistency in performance, even with a model 5-fold larger than vanilla ABMIL (5M parameters vs. 1M parameters), suggests that pancancer pretrained models are less likely to overfit. Furthermore, the monotonic increase in performance for PC-108 indicates that an effectively-initialized model may even exhibit favorable scaling properties, highlighting the potential and importance for large-scale slide-level foundation models. We also examine different Transformer sizes in Table A3, finding similar trends in transferability and model size.

### 4.5. Can supervised MIL transfer close the gap with slide foundation models?

We conduct a head-to-head comparison of our PC-108 pretrained models' transferability against two state-of-the-art slide-level foundation models: CHIEF (Wang et al., 2024) and GigaPath (Xu et al., 2024a). CHIEF, an ABMIL-based model, was pretrained on 60,530 slides using supervised contrastive learning (Khosla et al., 2021) and CLIP (Radford et al., 2021) to detect cancer across 19 organ types.

GigaPath, a LongNet-based model (Ding et al., 2023), was pretrained on 171,189 WSIs via a self-supervised masked autoencoder approach, with a ViT patch encoder pretrained on over 1.3 million patches using DINOv2.

For a fair comparison, we benchmark GigaPath against an ABMIL model pretrained with PC-108 using GigaPath's ViT patch features. Similarly, CHIEF is benchmarked against an ABMIL model pretrained with PC-108 using CTransPath features. Transfer quality is assessed by comparing both KNN and finetuning performance.

Our results reported in **Table 2** indicate that, while all pretrained models outperform training from scratch, PC-108-based slide representations yielded superior KNN performance in 12/15 tasks against CHIEF (average increase: +5.9%) and 13/15 tasks against GigaPath (average increase: +9.7%). This improvement is significant because PC-108's pretraining set had no sample overlap with these downstream tasks, whereas CHIEF was pretrained on TCGA and PANDA, which cover 11 of the 15 tasks. This underscores the high generalizability of PC-108's frozen slide-level features from pancancer pretraining.

These advantages extended to finetuning. PC-108 pretrained representations led to better finetuning performance than CHIEF in 11/15 tasks (+2.2% average improvement) and GigaPath in 10/15 tasks (+0.8% average improvement). Crucially, PC-108 achieved this using a pretraining dataset that was merely 6.5% and 2.3% the size of CHIEF's and GigaPath's, respectively. As both CHIEF and our PC-108 model employ the same ABMIL architecture, we conclude that PC-108's superior transferability stems from its pretraining methodology rather than architectural differences.

Although CHIEF and our work both employ supervised pretraining, we hypothesize that this difference in performance may likely be a result of the complexity of the pretraining dataset. Specifically, CHIEF was pretrained on a binary task

*Table 2.* **Comparison of MIL transfer with slide foundation models.** KNN and finetuning performance for MIL transfer with slide foundation models (GigaPath, CHIEF) compared against an ABMIL model initialized with pancancer pretraining (PC-108) and random weights (Base). Evaluations are grouped by patch encoder (GigaPath ViT, CTransPath). Best results are bold, second best are underlined.

| | KNN | | | | | | Finetuning | | | | | |
| | GigaPath ViT | | | CTransPath | | | GigaPath ViT | | | CTransPath | | |
| Task | PC-108 | GigaPath | Base | PC-108 | CHIEF | Base | PC-108 | GigaPath | Base | PC-108 | CHIEF | Base |
|---|---|---|---|---|---|---|---|---|---|---|---|---|
| BRACS-C | **70.3**(4.6) | 47.8(3.3) | 44.7(3.8) | **66.5**(3.5) | **66.5**(3.3) | 49.9(3.9) | **75.0**(4.9) | 65.6(4.1) | 69.4(4.8) | 71.8(5.1) | **74.6**(4.4) | 60.6(5.2) |
| BRACS-F | **35.3**(4.1) | 24.5(4.5) | 18.2(4.8) | **46.3**(4.2) | 37.5(4.8) | 16.4(5.1) | **48.3**(4.4) | 43.5(4.5) | 44.1(4.5) | **48.8**(5.1) | 42.1(5.1) | 43.1(4.0) |
| BRCA-ER | **88.4**(2.8) | 71.9(4.1) | 70.2(4.7) | **84.2**(3.8) | 77.4(3.5) | 69.1(3.9) | 84.8(4.4) | 81.7(4.4) | **86.9**(4.5) | **86.2**(3.7) | 84.0(4.0) | 85.3(3.6) |
| BRCA-HER2 | 62.7(5.0) | **65.9**(4.1) | 59.6(4.6) | 60.3(4.8) | 54.0(4.1) | **63.3**(4.0) | **68.9**(5.1) | 67.0(5.4) | 63.5(6.0) | 65.3(5.7) | 63.1(5.1) | **69.6**(4.9) |
| BRCA-PIK3CA | **60.8**(3.3) | 60.2(3.5) | 54.7(3.8) | **61.0**(3.5) | 60.6(3.7) | 52.9(3.7) | **66.9**(4.2) | 64.7(3.9) | 60.9(3.6) | **67.1**(4.3) | 62.9(3.9) | 55.7(4.1) |
| BRCA-PR | **83.0**(3.8) | 67.4(3.0) | 68.2(3.4) | 70.4(3.9) | **74.2**(3.6) | 67.5(4.1) | 75.3(3.6) | **78.6**(3.6) | 80.0(3.4) | **78.1**(3.4) | 75.5(3.7) | 75.5(4.0) |
| NSCLC-EGFR | **72.6**(7.2) | 37.0(7.0) | 58.8(7.4) | **71.1**(7.5) | 59.7(7.3) | 50.1(8.1) | **73.6**(7.6) | 58.1(8.1) | 62.3(6.7) | 58.8(9.9) | **65.1**(8.8) | 53.9(9.1) |
| NSCLC-KRAS | 60.8(4.2) | **63.2**(6.2) | 57.3(4.7) | 60.6(6.5) | 51.4(5.9) | 50.2(6.3) | 69.8(5.7) | **71.8**(5.4) | 58.0(5.2) | 69.1(6.2) | 57.5(6.3) | 67.1(5.7) |
| NSCLC-STK11 | **84.5**(5.1) | 77.4(6.7) | 63.2(5.7) | 64.5(5.9) | 62.3(5.9) | 50.1(6.2) | **88.6**(4.5) | 78.3(4.4) | 83.5(3.9) | 77.5(7.0) | 71.1(6.3) | 76.2(7.5) |
| NSCLC-TP53 | **80.3**(5.1) | 69.7(4.1) | 68.3(4.4) | 74.9(4.9) | 72.0(4.4) | 60.0(4.4) | 80.9(4.3) | 78.4(4.7) | 77.3(4.0) | **81.8**(4.3) | 79.8(4.6) | 75.2(4.8) |
| EBRAINS-C | **78.5**(2.3) | 72.9(2.0) | 74.6(2.3) | 67.2(2.1) | 51.8(2.2) | 58.1(2.5) | 85.5(2.2) | **86.4**(2.2) | 85.9(2.2) | 78.2(2.6) | **81.9**(2.3) | 79.5(2.2) |
| EBRAINS-F | **60.6**(2.2) | 56.5(1.9) | 54.2(1.7) | 51.9(1.8) | 38.6(1.5) | 37.1(2.3) | 70.2(2.0) | **72.1**(1.9) | 69.5(1.8) | 59.2(2.3) | **60.0**(2.6) | 58.3(2.2) |
| GBMLGG-C | **93.9**(0.6) | 84.4(1.4) | 79.7(1.2) | 89.4(0.9) | 77.0(0.8) | 82.3(1.1) | 93.4(2.1) | **94.7**(1.3) | 92.0(1.6) | 92.6(1.7) | 91.2(1.8) | 91.3(1.8) |
| GBMLGG-F | **55.3**(1.8) | 45.3(2.2) | 29.1(2.5) | 49.2(2.3) | 38.9(2.1) | 38.2(2.6) | 56.8(2.8) | 53.0(3.4) | 48.0(3.2) | **56.5**(3.3) | 54.4(3.3) | 56.4(3.5) |
| PANDA | 67.5(0.9) | **67.4**(1.0) | 53.0(0.9) | 75.0(1.1) | **83.2**(0.8) | 71.8(1.1) | **93.1**(1.1) | **94.5**(0.7) | 91.5(0.9) | **90.8**(1.0) | 84.2(1.4) | 90.5(1.0) |
| Average | **70.3** | 60.8 | 56.9 | **66.2** | 60.3 | 54.5 | **75.4** | 72.6 | 71.5 | **72.0** | 69.8 | 69.2 |

of distinguishing cancer from non-cancer, utilizing both supervised contrastive learning for visual pretraining and CLIP for image-text alignment. We believe the strength of our approach lies in the diversity of our pretraining dataset and the challenging nature of differentiating a large number of classes simultaneously: PC-108 is a pancancer, hierarchical classification scheme, requiring the model to implicitly characterize each WSI in terms of its organ, cancer type, and cancer subtype(s). This diverse and challenging pretraining task likely promotes the learning of comparatively more detailed, generalizable slide-level representations. Overall, our results highlight the feasibility of supervised pancancer pretrained models to surpass the ability of self-supervised slide-level foundation models while requiring less than 10% of the pretraining data.

### 4.6. How do patch encoders impact transfer performance?

To investigate whether the observed benefits of pancancer pretraining are inherent to MIL architecture versus choice of pretrained patch encoder, we also investigate transfer performance across various feature encoders of varying strengths. Specifically, we conduct experiments with a general-purpose image encoder for natural images, ResNet-50 (He et al., 2015) pretrained on IN-1k, CTransPath (Wang et al., 2022), GigaPath ViT (Xu et al., 2024b), UNIv2-h (Chen et al., 2024a) and CONCHv1.5 (Lu et al., 2024). For each encoder, we evaluate the finetuning transfer of randomly initialized ABMIL models to their counterparts initialized from PC-108-trained models.

The results of these experiments are displayed in **Tables 2 and 3**. We observe that across all encoders, PC-108 pretraining leads to an average improvement in performance.

The benefit of PC-108 pretraining is particularly evident for ABMIL, which improves over random initialization on 13/15 tasks with CTransPath, 12/15 tasks with ResNet50, 12/15 tasks with GigaPath, 10/15 tasks with UNIv2, and 11/15 tasks with CONCHv15. We repeat this experiment over CTransPath and ResNet50 with TransMIL (**Table A5**), finding that TransMIL leads to improved performance in 8/15 tasks with CTransPath and 10/15 tasks with ResNet.

*Table 3.* **Combined ABMIL pretraining performance with different encoders.** Performance across different tasks for ABMIL using ResNet-50, UNIv2, and CONCHv1.5 as patch feature encoders with PC-108 pretraining and random initialization (Base). Best performance between Base and PC-108 for each encoder is **bold**.

| | ResNet-50 | | UNIv2 | | CONCHv1.5 | |
| Task | Base | PC-108 | Base | PC-108 | Base | PC-108 |
|---|---|---|---|---|---|---|
| BRACS-C | 54.0 (4.9) | **54.1**(5.0) | 71.6(4.4) | **74.6**(4.8) | **78.4**(4.6) | 78.0(4.8) |
| BRACS-F | 19.4(3.6) | **27.3**(3.7) | 45.6(4.9) | **50.0**(4.9) | 49.4(3.5) | **53.2**(4.6) |
| BRCA-ER | **77.5**(4.0) | 72.3(5.1) | **85.3**(3.6) | 84.5(4.0) | 87.3(3.9) | **88.7**(4.0) |
| BRCA-HER2 | 58.0(6.3) | **65.5**(6.6) | **71.1**(5.3) | 67.1(5.9) | **64.5**(5.5) | 56.8(5.5) |
| BRCA-PIK3CA | 59.0(4.5) | **60.7**(4.0) | **62.7**(4.8) | 57.1(3.7) | 66.1(4.1) | **69.1**(3.6) |
| BRCA-PR | 65.6(4.4) | 62.2(4.8) | 76.4(3.6) | **77.7**(4.2) | 73.0(3.9) | **77.5**(3.9) |
| EBRAINS-C | 51.3(2.7) | **53.6**(2.8) | 89.3(2.1) | **90.5**(1.7) | **90.1**(1.9) | 88.0(2.2) |
| EBRAINS-F | 28.2(1.9) | **34.7**(2.0) | **70.6**(2.1) | 70.4(1.9) | 70.1(1.9) | **71.1**(2.0) |
| GBMLGG-C | 78.5(2.2) | **84.8**(2.3) | 91.0(2.1) | **92.7**(1.7) | 91.5(1.7) | **92.8**(1.6) |
| GBMLGG-F | 37.7(3.7) | **42.6**(3.8) | 56.2(3.2) | **59.2**(3.2) | 59.7(3.0) | **60.6**(2.4) |
| NSCLC-EGFR | **56.1**(9.7) | 52.0(9.8) | 70.9(10.1) | **72.4**(8.9) | 56.4(8.8) | **71.4**(7.0) |
| NSCLC-KRAS | 56.8(5.7) | **63.5**(5.8) | **58.9**(6.2) | 58.3(6.7) | 59.5(5.7) | **60.7**(6.6) |
| NSCLC-STK11 | 50.7(8.0) | **64.1**(8.1) | 70.4(8.3) | **87.9**(4.2) | 70.0(7.0) | **76.1**(7.6) |
| NSCLC-TP53 | 69.9(4.9) | **76.3**(5.0) | 81.0(4.9) | **83.1**(4.5) | 76.2(4.9) | **83.1**(4.8) |
| PANDA | 79.6(1.6) | **80.1**(1.7) | 93.3(0.7) | **93.8**(0.7) | **91.8**(0.8) | 91.4(0.9) |
| Average | 56.2(4.5) | **59.6**(4.7) | 72.9(4.4) | **74.6**(4.1) | 72.3(4.1) | **74.6**(4.1) |

### 4.7. Can public, single-organ datasets serve as a pretraining task?

Section 4.1 establishes that pancancer pretraining produces transferable frozen slide-level representations, although single-organ pretraining also consistently outperforms ran-

dom weight initialization. Here, we investigate if these findings extend to finetuning. To compare the benefits of pancancer and single-organ pretraining, we evaluate the transferability of four models (ABMIL, DFTD, TransMIL, and Transformer). We train each model on a specific task from our 19 public datasets and then use its weights to initialize the model for evaluation on all other tasks.

The results, summarized in **Figure A1**, align with our previous findings on frozen feature evaluation. We observe that any form of pretraining improves average downstream task performance over random initialization, even when transferring between different organ types. Furthermore, PC-108 consistently achieves the highest finetuning performance, surpassing all single-organ pretraining approaches. These findings highlight pancancer pretraining as the preferable strategy for developing generalizable models with supervised pretraining, while validating single-organ pretraining as a feasible and accessible alternative.

### 4.8. Why does pancancer pretraining transfer so well?

Encouraged by the performance of pancancer pretrained models, we next seek to understand why initialization from pancancer tasks would lead to higher performance with finetuning. We hypothesize that, because large pancancer tasks require models to learn slide-level representations which can distinguish between a huge variety of organs and morphological appearances, these slide-level representations would likely help differentiate between classes of unseen tasks without any finetuning. In **Figure 5**, we indeed observe clearer class separation among 12 histologic subtypes (EBRAINS-C) for slide features from ABMIL trained on PC-108, when compared to randomly initialized version.

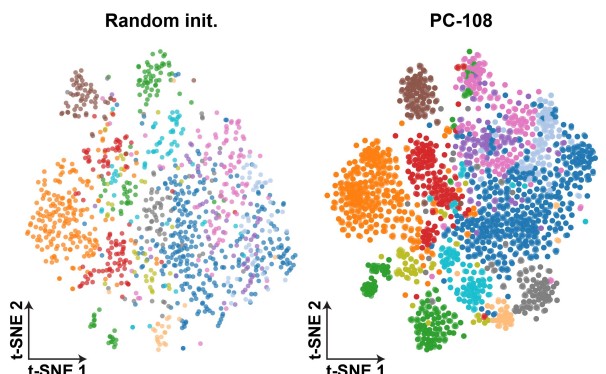

*Figure 5.* **t-SNE of slide-level features**. Visualization of the slide-level features from randomly initialized ABMIL compared to ABMIL pretrained on PC-108 for 12-class brain subtyping for rare brain disease classification.

### 4.9. What features are being transferred?

Supervised MIL transfer learning offers substantial performance gains, but it remains unclear which layer's features are most important for transfer. While early layers often prove most transferable in standard CNNs (Raghu et al., 2019), this principle may not extend to MIL models. These models, designed for gigapixel inputs, use a distinct two-stage process, consisting of a patch-level MLP followed by attention-based aggregation, raising the questions: How much do pretrained layers change during finetuning, and how does their transfer impact performance? We examine quantitative and qualitative measures to answer these questions in this section.

#### 4.9.1. FEATURE STABILITY

We first study feature stability of pretrained layers, hypothesizing that layers possessing the most transferable knowledge would change less during fine-tuning. We measure feature stability with Singular Vector Canonical Correlation Analysis (SVCCA) (Raghu et al., 2017), which provides a means of assessing the linear similarity between the activation spaces of each neural network layer before and after finetuning. We measure layer stability with using the average canonical correlation on a scale from 0-100.

We compare the pre- and post-fine-tuning activations for every layer in our Standard (S, $9 \times 10^5$ params) and Large (L, $5.25 \times 10^6$ params) ABMIL models, using 45,232 patches from 30 NSCLC-KRAS slides. ABMIL-S uses a single linear layer (Lin. 1) for patch-specific processing, while ABMIL-L uses a three-layer MLP (Lin. 1, 2, 3) and wider aggregation layer, allowing us to investigate whether the same trends hold across different model scales.

The results, presented in **Table 4**, demonstrate that pretrained layers exhibit significantly less change during training compared to their randomly-initialized counterparts. Notably, the deep attention layer shows a remarkably low correlation ($2.5 \pm 0.0$ and $16.3 \pm 0.0$ for ABMIL-L and -S, respectively) with its initial state when training from random initialization. In contrast, when using pretrained weights, these models retain high similarity ($82.7 \pm 0.0$ and $97.7 \pm 0.0$). Given that the attention layer plays the critical role of converting each patch embedding into a scalar weight for slide-level aggregation, this finding strongly suggests that the benefits of MIL transfer learning are closely tied to the transfer of learned aggregation strategies.

#### 4.9.2. LAYER-LEVEL PRETRAINING IMPORTANCE

We next investigate the contribution of different components within the pretrained MIL model towards overall performance. To do so, we assess finetuning performance while progressively resetting layers of the aggregation module to

*Table 4.* **SVCCA Similarity.** Mean and standard deviation of canonical correlation between layers before and after fine-tuning, on the scale of 0 to 100 for Large (L) and Standard (S) ABMIL.

| Layer | Base-L | PC108-L | Base-S | PC108-S |
|---|---|---|---|---|
| Lin. 1 | $92.8 \pm 18.3$ | $95.5 \pm 14.7$ | $93.9 \pm 16.0$ | $97.2 \pm 12.0$ |
| Lin. 2 | $81.2 \pm 22.6$ | $89.2 \pm 17.9$ | — | — |
| Lin. 3 | $47.2 \pm 25.0$ | $66.5 \pm 26.3$ | — | — |
| Attention | $2.5 \pm 0.0$ | $82.7 \pm 0.0$ | $16.3 \pm 0.0$ | $97.7 \pm 0.0$ |
| Average | 61.6 | **83.1** | 75.7 | **96.2** |

random weights, starting from the final layer and moving to earlier layers. We compare these results against the baseline where all weights are transferred for PC108-pretrained ABMIL-L (PC108-L).

The experiments are setup as follows - **Reset Attn**: Re-initialize only the attention layer weights. **Reset Lin3+**: Re-initialize the third linear layer and attention layer. **Reset Lin2+**: Re-initialize the second and third linear layers, and the attention layer. **Reset All**: Re-initialize all weights in the aggregation module, same as randomly initialized weights.

*Table 5.* **Impact of re-initializing aggregation module layers**. Values represent the change in performance relative to the PC108-L baseline (full weight transfer).

| Task | PC108-L | Reset Attn | Reset Lin3+ | Reset Lin2+ | Reset All |
|---|---|---|---|---|---|
| BRACS-C | 71.9 | -8.5 | -8.5 | -12.8 | -11.0 |
| BRACS-F | 53.3 | -12.1 | -10.5 | -10.4 | -10.1 |
| GBMLGG-C | 95.4 | -1.2 | -1.0 | 0.0 | -0.2 |
| GBMLGG-F | 51.7 | 0.0 | -0.8 | -1.8 | 0.8 |
| NSCLC-EGFR | 76.1 | -8.6 | -10.2 | -13.1 | -12.8 |
| NSCLC-KRAS | 68.4 | -2.0 | -4.5 | -5.8 | -8.7 |
| NSCLC-STK11 | 86.7 | 0.0 | -0.6 | -8.3 | -15.0 |
| NSCLC-TP53 | 81.5 | -7.3 | -5.9 | -0.5 | -10.0 |
| Average | 73.1 | -5.0 | -5.2 | -6.6 | -8.3 |

Performance is reported as the change relative to the full weight transfer (PC108) baseline. The results, presented in **Table 5**, reveal that re-initializing the attention layer (Reset Attn) causes the largest single performance drop (-5.0% average decrease). Re-initializing the preceding MLP layers (Reset Lin3+ and Reset Lin2+) leads to further, albeit smaller, decreases, culminating in an 8.3-point drop when all weights are reset. This underscores the critical role of all MIL layers, particularly the attention component, for successful transfer learning. These findings diverge from prior work (Raghu et al., 2017), which observed minimal impact from transferring later CNN layers, thereby highlighting the unique characteristics and requirements of MIL transfer.

### 4.9.3. VISUALIZING TRANSFERABILITY

We examine whether the transferability observed in **Sections 4.9.1 and 4.9.2** is also reflected in ABMIL attention heatmaps, which indicate the learned importance of each patch according to the aggregation layer. We show

a representative attention heatmap of TP53 positive lung cancer (TCGA) in **Figure 6** across three different scenarios: Randomly-initialized weights, PC-108 pretrained weights, and finetuned on NSCLC TP53 mutation prediction from PC-108 pretrained weights.

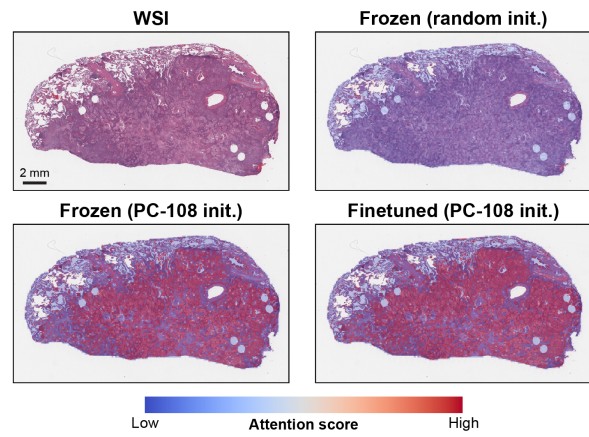

*Figure 6.* **Heatmaps for visualizing attention transfer**. Visualization of the three different ABMIL attention heatmaps for lung squamous cell carcinoma: ABMIL with randomly-initialized weights, PC-108 pretrained weights, and finetuned with NSCLC TP53 mutation prediction from PC-108 pretrained weights.

We observe that ABMIL with random weights exhibit diffuse attention, without focusing on regions of clinical importance. Meanwhile, the pancancer pretrained model focuses on tumor regions prior to finetuning, with high-attention regions remaining relatively stable after finetuning. We hypothesize that by allowing the model to place high attention from the start on regions that are diagnostically-relevant across cancer types, pretrained MIL models may be less prone to spurious correlations and learn more nuanced distinctions between classes over the course of training.

## 5. Conclusion

We investigated transfer learning capabilities of MIL models in CPath, with models pretrained on pancancer tasks exhibiting remarkable generalization across organs and tasks, and models pretrained on single-organ tasks transferring effectively to out-of-domain tasks. Limitations of our study include the absence of state-space MIL models, survival prediction tasks, and augmented pretraining that could further improve generalizability. Nonetheless, by providing insights into MIL transfer, this work offers a roadmap for leveraging supervised pretrained MIL models to enhance performance in various clinical tasks. Model weights for our best-performing ABMIL model, named FEATHER, as well as initialization schemes for all MIL methods, can be accessed at https://github.com/mahmoodlab/MIL-Lab.

## Impact Statement

The research presented in this paper aims to advance the application of Machine Learning in computational pathology by introducing a more resource-efficient paradigm for pretraining multiple instance learning models. Our work demonstrates that leveraging an institution's own diverse, pancancercer data can lead to significant improvements in data efficiency, requiring less data than traditional large-scale approaches. This, in turn, contributes to notable compute and storage efficiency for both model development and subsequent fine-tuning. The primary societal benefit of this approach is the potential to make advanced AI tools for pathology more accessible, particularly for research groups and healthcare institutions that may lack the extensive resources typically required for pretraining with techniques like large-scale self-supervised learning (SSL). To this end, we share the model weights for our best-performing MIL pretrained model, an ABMIL model which we call FEATHER, and initialization methods in the aforementioned GitHub for public use. We envision this fostering wider innovation, accelerating the development of diagnostic aids, and potentially improving the consistency and reach of pathological services by making powerful AI tools easier to develop and adapt.

While this work offers a path toward more accessible and efficient AI development, it is crucial to address potential ethical considerations and societal consequences:

**Algorithmic Bias and Equity**: Models pretrained on data from a specific institution, even if pancancercer, may inadvertently inherit biases present in that data (e.g., related to patient demographics, disease prevalence, specific equipment, or institutional recording practices). In the case of PC-108, all cases were acquired from Brigham and Women's Hospital, which may lead to bias and performance disparities when applied to underrepresented groups or different clinical contexts, potentially exacerbating existing health inequities. In the future, more rigorous bias audits, diverse dataset validation, and ongoing monitoring are therefore critical next steps to ensure the equitable translation of pretrained models.

**Resource Allocation and Access**: While our approach lowers the barrier for model development, ensuring equitable access to the benefits of these AI tools across diverse healthcare settings, including those in low-resource environments, will require concerted effort beyond the technical aspects of model creation.

Overall, our work contributes to a more sustainable, efficient, and potentially equitable approach to developing AI tools in computational pathology. The realization of these benefits, however, depends critically on a continued commitment to responsible research and development practices.

This includes transparency in model development and limitations, proactive strategies to identify and mitigate algorithmic bias, and ongoing efforts to ensure that the advantages of these technologies are accessible equitably across diverse healthcare settings.

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

## A. Transfer performance across pretraining tasks

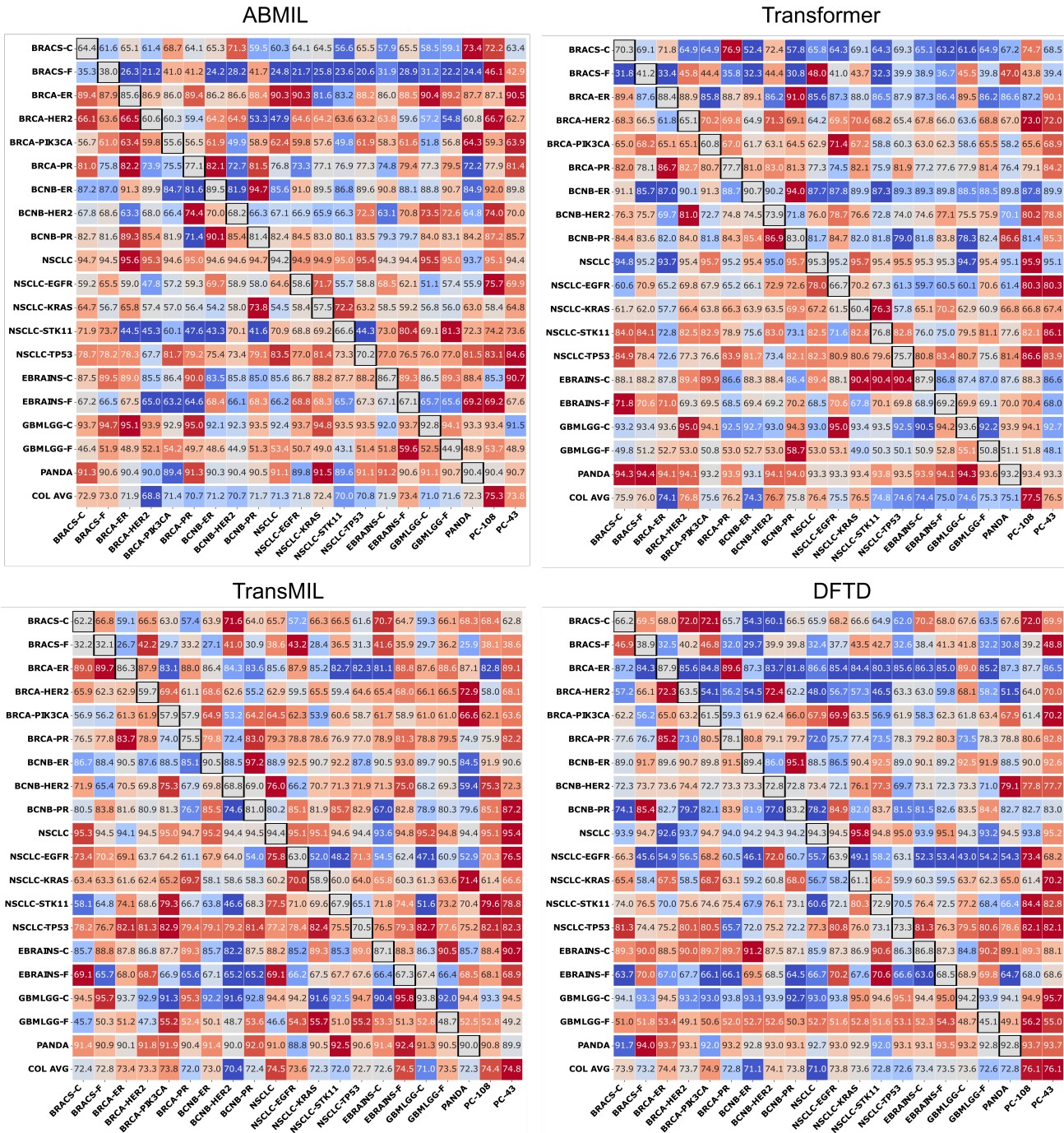

*Figure A1.* **Finetuning performance with pretraining on public datasets.** Performance of various MIL models initialized with different pretraining tasks (columns) on each evaluation dataset (rows). Diagonals indicate performance when trained from scratch. Colormap is row-normalized according to the difference between each pretrained model and the model trained from scratch, such that red corresponds to an increase in performance compared to random initialization, and blue indicates a decrease in performance. Metrics reported are AUROC for binary classification tasks, weighted Cohen's Kappa for PANDA, and balanced accuracy for all other multiclass problems. We find that across all models and pretraining tasks, pretraining generally leads to an improvement in performance (indicated by the large proportion of red). Each model was finetuned for 10 epochs.

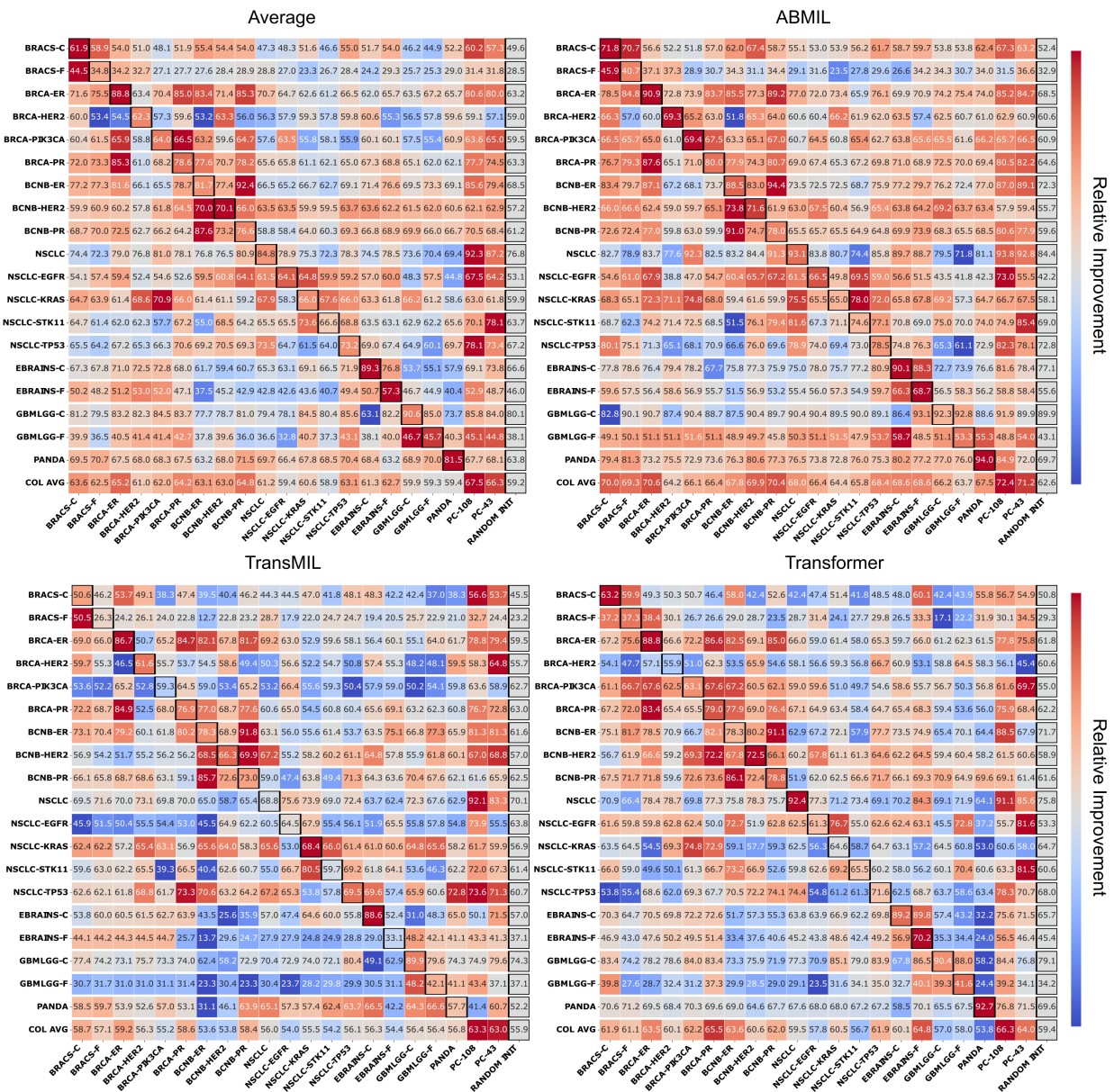

Figure A2. **KNN performance on public datasets.** Performance of various MIL models initialized with different pretraining tasks (columns) on each evaluation dataset (rows). Colormap is row-normalized according to maximum and minimum performance on each task. Metrics reported are AUROC for binary classification tasks, weighted Cohen's Kappa for PANDA, and balanced accuracy for all other multiclass problems. We find that across transformer-based models, the PC-108 pretrained model consistently resulted in the best knn transfer across pretraining tasks. Meanwhile, the best-performing pretraining model varied.

# B. Implementation details

### B.1. Multiple Instance Learning implementation

All MIL models are adapted according to their official implementation, using the default hyperparameters provided by their official codebases. For meanMIL and maxMIL, which do not have official codebases, we first apply a linear layer paramaterized by $W \in \mathbb{R}^{d \times 512}$, followed by ReLU activation, onto each $d$-dimensional patch embedding. We then obtain a slide-level prediction by feeding the average of task-specific embeddings through a classification head. For **MaxMIL**, we feed each patch through $W$, then pass each transformed patch embedding through a classification head, and select the patch with the highest logit as the final slide representation.

### B.2. Training details

We train all models with the AdamW optimizer with a learning rate of $1 \times 10^{-4}$, a cosine decay scheduler, and mixed precision according to Pytorch's native implementation. For datasets with a validation set, we train with a maximum of 20 epochs with an early stopping patience of 5 epochs for a minimum of 10 epochs. For datasets without a validation set, we train for 10 epochs. We use cross-entropy loss with random class-weighted sampling and a batch size of 1. For regularization, we use a weight decay of $1 \times 10^{-5}$, a dropout of 0.25 at every feedforward layer, and a dropout of 0.1 on the features from the pretrained encoder. Experiments were performed across four NVIDIA RTX A4000s, three NVIDIA GeForce RTX 2080 TIs GPUs, and three RTX 3090s, with a single GPU used per experiment.

# C. Dataset Description

We briefly describe the datasets that were used to evaluate MIL transfer.

### C.1. Morphological Subtyping

**EBRAINS** (Roetzer-Pejrimovsky et al., 2022): We perform coarse-grained (12 classes) and fine-grained (30 classes) classification of brain tumor subtypes. The dataset consisted 2,319 Hematoxylin and Eosin (H&E) Formalin-fixed and paraffin-embedded (FFPE) Whole Slide Images (WSIs). We use label-stratified train/val/test splits (50% / 25% / 25%) provided by UNI(Chen et al., 2024a) with the same folds for both coarse- and fine-grained tasks. We evaluate performance using balanced accuracy.

**NSCLC**: The non-small cell lung carcinoma (NSCLC) subtyping task was a binary classification problem for distinguishing lung adenocarcinoma (LUAD) and lung squamous cell carcinoma (LUSC). The training data consisted of publicly available H&E WSIs from TCGA ($n = 1,041$ slides). We performed an 80% / 10% / 10% train/val/test split on the TCGA dataset for training and internal validation, and evaluated the trained model on two external datasets: the Clinical Proteomic Tumor Analysis Consortium (CPTAC, $n = 1,091$ slides) and the National Lung Screening Trial (NLST, $n = 1,008$ slides) (Campbell et al., 2016; Satpathy et al., 2021; Gillette et al., 2020). We report the AUROC averaged across the TCGA, NLST, and CPTAC test sets for all results.

**PANDA** (Bulten et al., 2022): We used prostate cancer core needle biopsies ($n = 10,616$) from the Prostate Cancer Grade Assessment (PANDA) challenge to perform 6-class classification according to the prostate cancer grade. We use the same train/val/test folds (80% / 10 % / 10%) as UNI, and evaluate using Cohen's quadratic weighted Kappa $\kappa$ metric.

**BRACS** (Brancati et al., 2021): The BRACS subtyping task consisted of a 3-class coarse-grained classification task to distinguish benign, malignant, and atypical breast carcinoma H&E slides, as well as a fine-grained 7-class classification task that classifies benign tumors into three subtypes, atypical tumors into two subtypes, and malignant tumors as two subtypes. We use the official train/val/test folds (72% / 12% / 16%), with the same folds for both coarse- and fine-grained tasks. We evaluate performance using balanced accuracy.

**Pancancer subtyping**: These tasks consisted of cases collected at Brigham and Women's Hospital, from which we defined a 108-class fine-grained label and a 43-class coarse-grained label set. The fine-grained task sought to identify one of 108 OncoTree codes, while PC-43 sought to identify one of 43 cancer types. The cases were divided into a ratio of 3,944 slides for training, and 1,620 slides for validation, with the same splits for both tasks. These datasets were previously named OT-108 and OT-43 in the UNI study (Chen et al., 2024a).

## C.2. Biomarker Prediction

**Lung cancer biomarkers**: We evaluate H&E-stained WSIs for the binary classification task of predicting mutation status of TP53, KRAS, STK11, and EGFR in TCGA lung cancer cases ($n = 524$ slides) (Network et al., 2015), with each task site- and label-stratified into an approximate train/val/test splits (60% / 20% / 20%), with the same splits for all tasks. We evaluate performance using AUROC.

**Breast cancer biomarkers**: We predicted the binary mutation status of ER, PR, HER2, and PIK3CA on H&E-stained WSIs from TCGA breast cancer (BRCA) cases ($n = 1,034$), each site-stratified and label-stratified in an approximate train/val/test splits (60% / 20% / 20%). Additionally, we evaluate on breast cancer core needle biopsies (BCNB, $n = 1,058$) (Xu et al., 2021) in a label-stratified train/test split (90% / 10%) for ER, PR, and HER2, with the same splits for all tasks. We evaluate performance using AUROC.

**GBMLGG mutational subtyping** (Brennan et al., 2013; Roetzer-Pejrimovsky et al., 2022): These tasks include binary coarse-grained mutation prediction of IDH1 status using the TCGA GBMLGG dataset (1,123 slides), and 5-class fine-grained histomolecular subtyping. The 5-class histomolecular subtyping task was separated into the categories of Astrocytoma, IDH1-mutant, Glioblastoma, IDH1-mutant, Oligodendroglioma, IDH1-mutant and 1p/19q codeleted, Astrocytoma, IDH1-wildtype, and Glioblastoma, IDH1-wildtype. For training and evaluation of both tasks, we use the UNI splits, which label-stratified TCGA-GBMLGG into a train/val/test split with a 47:22:31 ratio with the same splits for both coarse- and fine-grained tasks. Additionally, we perform external validation on the held-out EBRAINS cohort ($n = 873$ slides) for the cases with known IHD1 status. We evaluate GBMLGG-coarse with AUROC, and GBMLGG-fine with balanced accuracy.

**Shared split experiments**: For the experiments depicted in Figures A1 and A2, we perform transfer learning by pretraining on the training set of one dataset and evaluating on the test set of a separate dataset. However, we observed that some official dataset splits (specifically, NSCLC, BRCA, and BCNB) contain overlapping samples—meaning that certain instances appear in the training set for one label and the test set for another label. This overlap could result in data leakage during evaluation, thereby inflating performance estimates. To mitigate this risk, we employ multi-label stratified splits for the NSCLC, BRCA, and BCNB datasets in these specific experiments. With this approach, we ensure that all tasks within a dataset (e.g., NSCLC TP53 and NSCLC EGFR) share an identical set of training samples, eliminating any possibility of overlap between training and testing data across labels. These custom multi-label splits are publicly available at the GitHub repository. For all other experiments, we use the official or previously published dataset splits. This decision facilitates reproducibility and enables direct comparison with prior work.

# D. Expanded Results

## D.1. ABMIL Scaling Implementation

*Table A1.* Model Configurations for ABMIL with different parameter counts with UNI patch features. Embed dim: output dimension of pre-attention linear layer. Attn Dim: Hidden dimension of attention2 embedding. Num FC Layers: number of pre-attention linear layers. FC Hidden Dim: hidden dimension(s) of pre-attention linear layers.

| Params | In Dim | N Classes | Embed Dim | Attn Dim | Num FC Layers | Hidden Dim |
|---|---|---|---|---|---|---|
| 8,530,675 | 1,024 | 2 | 512 | 512 | 5 | [2048, 1536, 1024, 768] |
| 6,837,292 | 1,024 | 2 | 512 | 512 | 4 | [2048, 1280, 768] |
| 5,249,027 | 1,024 | 2 | 512 | 512 | 3 | [2048, 1024] |
| 3,084,931 | 1,024 | 2 | 512 | 384 | 3 | [1280, 768] |
| 1,445,507 | 1,024 | 2 | 512 | 384 | 3 | [512, 512] |
| 920,195 | 1,024 | 2 | 512 | 384 | 1 | [] |
| 591,747 | 1,024 | 2 | 384 | 256 | 1 | [] |
| 394,755 | 1,024 | 2 | 256 | 256 | 1 | [] |
| 164,611 | 1,024 | 2 | 128 | 128 | 1 | [] |

*Table A2.* Model Configurations for Transformer with different parameter counts with 1024-dimensional UNI patch features. N. Layers: Number of Transformer encoder layers. Encoder Hidden Dim: Hidden dimension of encoder feedforward network. N. FC Layers: number of pre-attention linear layers. FC Hidden Dim: hidden dimension(s) of pre-attention linear layers. "-" indicates no feedforward network.

| Params | Embed Dim | N. Layers | Encoder Hidden Dim | N. FC Layers | FC Hidden Dim |
|---|---|---|---|---|---|
| 9,984,514 | 512 | 3 | 2048 | 1 | [] |
| 6,837,555 | 512 | 3 | 1024 | 1 | [] |
| 5,258,242 | 512 | 2 | 1024 | 3 | [512, 512] |
| 3,155,970 | 512 | 2 | - | 3 | [1280, 768] |
| 2,631,209 | 512 | 2 | - | 1 | [] |
| 792,316 | 256 | 2 | - | 1 | [] |
| 264,450 | 128 | 2 | - | 1 | [] |

*Table A3.* Performance comparison of ABMIL and Transformer models across a range of sizes, both with pretraining (PC-108) and without pretraining (Base). Performance is averaged across eight tasks: BRCA ER/PR, NSCLC TP53/STK11, GBMLGG C/F, and BRACS C/F. The results show that ABMIL performance plateaus in the 5-7M parameter range before decreasing at 9M. At larger sizes, Transformers benefit substantially from pretraining, outperforming ABMIL at 7M, though both models show reduced performance at 9M.

| Approx Params | Base | | PC108 | |
|---|---|---|---|---|
| | ABMIL | Transformer | ABMIL | Transformer |
| 0.2M | 70.5 | 67.4 | 73.1 | 72.9 |
| 1M | 70.1 | 68.8 | 72.6 | 72.6 |
| 2.6M | - | 64.4 | - | 70.5 |
| 3.2M | 71.3 | 65.2 | 74.0 | 71.2 |
| 5.2M | 70.7 | 65.1 | 75.7 | 73.8 |
| 7M | 71.3 | 66.3 | 75.4 | 76.8 |
| 9M | 70.6 | 67.5 | 74.1 | 72.1 |

*Table A4.* **Performance across random seeds.** Average performance with 95% confidence interval according to five runs with different random seeds. For each task, base models were randomly initialized with different weights for each random seed. PC-108 models used the same pretrained model and were finetuned with 5 random seeds for each downstream task. Average performance and average of standard deviation is shown in the final row and final column.

| Task | Initialization | ABMIL | DFTD | RRT | TransMIL |
|---|---|---|---|---|---|
| BRACS-C | Base | $71.4 \pm 2.7$ | $64.6 \pm 5.4$ | $65.7 \pm 3.7$ | $63.9 \pm 6.5$ |
|  | PC-108 | $68.8 \pm 2.4$ | $63.7 \pm 5.4$ | $68.6 \pm 0.0$ | $63.6 \pm 2.0$ |
|  | $\Delta$ | $-2.6$ | $-1.0$ | $+2.8$ | $-0.3$ |
| BRACS-F | Base | $41.3 \pm 2.3$ | $44.6 \pm 1.9$ | $35.4 \pm 2.8$ | $34.6 \pm 3.1$ |
|  | PC-108 | $40.9 \pm 1.8$ | $42.0 \pm 4.6$ | $36.2 \pm 1.3$ | $40.0 \pm 2.2$ |
|  | $\Delta$ | $-0.4$ | $-2.6$ | $+0.9$ | $+5.4$ |
| NSCLC Morph | Base | $95.4 \pm 0.3$ | $95.4 \pm 0.4$ | $94.4 \pm 0.6$ | $95.0 \pm 0.2$ |
|  | PC-108 | $95.2 \pm 0.1$ | $95.4 \pm 0.5$ | $94.7 \pm 0.2$ | $95.1 \pm 0.5$ |
|  | $\Delta$ | $-0.2$ | $+0.1$ | $+0.3$ | $+0.1$ |
| BCNB-ER | Base | $89.2 \pm 1.5$ | $91.1 \pm 0.2$ | $90.6 \pm 0.8$ | $90.5 \pm 0.2$ |
|  | PC-108 | $88.8 \pm 1.1$ | $90.1 \pm 1.6$ | $89.6 \pm 1.4$ | $88.7 \pm 1.3$ |
|  | $\Delta$ | $-0.4$ | $-1.0$ | $-1.0$ | $-1.8$ |
| BCNB-HER2 | Base | $74.5 \pm 1.1$ | $72.6 \pm 0.8$ | $70.7 \pm 1.1$ | $68.4 \pm 0.7$ |
|  | PC-108 | $75.5 \pm 2.5$ | $72.7 \pm 2.6$ | $72.6 \pm 1.3$ | $66.6 \pm 3.5$ |
|  | $\Delta$ | $+1.0$ | $+0.1$ | $+1.9$ | $-1.8$ |
| BCNB-PR | Base | $82.8 \pm 0.7$ | $83.0 \pm 0.4$ | $81.3 \pm 0.9$ | $80.6 \pm 0.3$ |
|  | PC-108 | $82.4 \pm 1.9$ | $82.6 \pm 2.1$ | $75.1 \pm 1.4$ | $81.6 \pm 5.1$ |
|  | $\Delta$ | $-0.4$ | $-0.4$ | $-6.2$ | $+1.0$ |
| BRCA-ER | Base | $87.4 \pm 1.0$ | $87.7 \pm 1.3$ | $83.5 \pm 1.4$ | $86.1 \pm 2.1$ |
|  | PC-108 | $87.5 \pm 1.3$ | $87.3 \pm 1.4$ | $86.3 \pm 1.3$ | $81.9 \pm 3.6$ |
|  | $\Delta$ | $+0.0$ | $-0.4$ | $+2.8$ | $-4.2$ |
| BRCA-HER2 | Base | $65.2 \pm 3.4$ | $63.6 \pm 3.3$ | $57.2 \pm 3.1$ | $61.1 \pm 3.4$ |
|  | PC-108 | $67.6 \pm 2.9$ | $65.7 \pm 4.0$ | $53.9 \pm 2.2$ | $61.7 \pm 4.4$ |
|  | $\Delta$ | $+2.3$ | $+2.1$ | $-3.3$ | $+0.6$ |
| BRCA-PIK3CA | Base | $61.6 \pm 1.8$ | $64.3 \pm 4.5$ | $60.2 \pm 2.6$ | $58.7 \pm 4.0$ |
|  | PC-108 | $66.8 \pm 1.4$ | $64.5 \pm 3.3$ | $60.0 \pm 3.1$ | $58.8 \pm 0.9$ |
|  | $\Delta$ | $+5.2$ | $+0.2$ | $-0.2$ | $+0.1$ |
| BRCA-PR | Base | $77.7 \pm 0.2$ | $79.5 \pm 2.1$ | $74.6 \pm 1.7$ | $76.7 \pm 2.3$ |
|  | PC-108 | $78.4 \pm 3.2$ | $79.5 \pm 1.4$ | $81.5 \pm 1.0$ | $79.6 \pm 2.2$ |
|  | $\Delta$ | $+0.7$ | $-0.1$ | $+6.9$ | $+2.9$ |
| EBRAINS-C | Base | $87.7 \pm 1.1$ | $84.4 \pm 1.7$ | $87.5 \pm 2.5$ | $85.6 \pm 0.8$ |
|  | PC-108 | $86.5 \pm 1.4$ | $88.2 \pm 1.8$ | $88.5 \pm 0.0$ | $87.4 \pm 2.8$ |
|  | $\Delta$ | $-1.2$ | $+3.7$ | $+1.0$ | $+1.8$ |
| EBRAINS-F | Base | $68.8 \pm 1.4$ | $65.9 \pm 0.8$ | $68.9 \pm 0.6$ | $67.4 \pm 1.0$ |
|  | PC-108 | $68.8 \pm 1.0$ | $66.8 \pm 1.6$ | $68.2 \pm 1.1$ | $67.3 \pm 1.4$ |
|  | $\Delta$ | $+0.0$ | $+0.8$ | $-0.8$ | $-0.1$ |
| GBMLGG-C | Base | $93.7 \pm 1.5$ | $92.5 \pm 1.1$ | $93.7 \pm 0.7$ | $92.8 \pm 1.3$ |
|  | PC-108 | $93.6 \pm 1.3$ | $91.7 \pm 1.1$ | $93.0 \pm 0.7$ | $93.5 \pm 0.5$ |
|  | $\Delta$ | $-0.1$ | $-0.7$ | $-0.7$ | $+0.6$ |
| GBMLGG-F | Base | $50.8 \pm 1.0$ | $46.9 \pm 4.3$ | $52.3 \pm 1.0$ | $49.0 \pm 2.4$ |
|  | PC-108 | $51.6 \pm 2.0$ | $51.8 \pm 2.7$ | $55.2 \pm 3.7$ | $49.9 \pm 3.4$ |
|  | $\Delta$ | $+0.8$ | $+4.9$ | $+2.9$ | $+0.9$ |
| NSCLC-EGFR | Base | $66.3 \pm 2.7$ | $70.1 \pm 3.3$ | $65.8 \pm 6.1$ | $62.7 \pm 1.5$ |
|  | PC-108 | $75.5 \pm 4.9$ | $71.7 \pm 4.9$ | $69.1 \pm 3.5$ | $69.9 \pm 4.3$ |
|  | $\Delta$ | $+9.2$ | $+1.6$ | $+3.3$ | $+7.2$ |
| NSCLC-KRAS | Base | $63.6 \pm 2.7$ | $62.3 \pm 4.4$ | $62.2 \pm 1.4$ | $61.7 \pm 4.4$ |
|  | PC-108 | $62.4 \pm 4.2$ | $61.8 \pm 3.1$ | $67.5 \pm 3.9$ | $63.3 \pm 3.1$ |
|  | $\Delta$ | $-1.2$ | $-0.4$ | $+5.3$ | $+1.6$ |
| NSCLC-STK11 | Base | $76.7 \pm 1.8$ | $78.3 \pm 2.9$ | $75.8 \pm 2.1$ | $62.4 \pm 7.6$ |
|  | PC-108 | $81.7 \pm 3.5$ | $80.4 \pm 2.6$ | $82.7 \pm 2.6$ | $75.3 \pm 3.5$ |
|  | $\Delta$ | $+5.0$ | $+2.1$ | $+6.9$ | $+12.9$ |
| NSCLC-TP53 | Base | $76.2 \pm 2.6$ | $80.9 \pm 3.9$ | $74.0 \pm 1.2$ | $74.0 \pm 2.6$ |
|  | PC-108 | $80.8 \pm 1.0$ | $80.5 \pm 2.9$ | $73.7 \pm 2.3$ | $79.4 \pm 2.2$ |
|  | $\Delta$ | $+4.6$ | $-0.4$ | $-0.2$ | $+5.4$ |
| PANDA | Base | $93.3 \pm 0.4$ | $93.1 \pm 0.5$ | $91.1 \pm 0.6$ | $90.5 \pm 0.4$ |
|  | PC-108 | $93.7 \pm 0.3$ | $92.8 \pm 0.5$ | $91.7 \pm 0.3$ | $90.7 \pm 0.4$ |
|  | $\Delta$ | $+0.4$ | $-0.3$ | $+0.6$ | $+0.3$ |
| Average | Base | $74.7 \pm 2.27$ | $74.8 \pm 2.14$ | $72.9 \pm 1.78$ | $70.9 \pm 2.22$ |
|  | PC-108 | $76.1 \pm 1.62$ | $75.2 \pm 1.98$ | $74.1 \pm 1.51$ | $73.4 \pm 2.14$ |
|  | $\Delta$ | $+1.4$ | $+0.4$ | $+1.2$ | $+2.6$ |

*Table A5.* **Pretraining performance with different encoders for TransMIL.** Performance across different tasks for TransMIL using CTransPath and ResNet as patch feature encoders with PC-108 pretraining and random initialization. Best performance between Base and PC-108 for each encoder is **bold**.

| Task | CTransPath | | ResNet | |
|---|---|---|---|---|
| | Base | PC-108 | Base | PC-108 |
| BRACS-C | **59.1** | 57.6 | 51.0 | **54.1** |
| BRACS-F | **39.0** | 38.6 | 26.1 | **29.9** |
| EBRAINS-C | **81.0** | 76.5 | **63.9** | 61.4 |
| EBRAINS-F | 53.9 | **54.9** | **46.8** | 42.5 |
| PANDA | **90.2** | 87.2 | **83.5** | 77.6 |
| BRCA-ER | **81.6** | 80.5 | **70.5** | 64.2 |
| BRCA-HER2 | **62.9** | 58.6 | **70.5** | 64.9 |
| BRCA-PIK3CA | 52.8 | **53.1** | 52.4 | **69.5** |
| BRCA-PR | 57.7 | **76.3** | 59.6 | **68.3** |
| NSCLC-EGFR | 59.1 | **69.4** | 48.2 | **66.0** |
| NSCLC-KRAS | 59.1 | **65.8** | 65.0 | **67.7** |
| NSCLC-STK11 | 62.4 | **79.0** | 64.3 | **76.9** |
| NSCLC-TP53 | 72.5 | **76.3** | 60.8 | **74.8** |
| GBMLGG-C | **91.4** | 89.6 | 53.4 | **78.7** |
| GBMLGG-F | 53.0 | **55.7** | 53.0 | **60.8** |
| Average | 65.1 | **67.94** | 59.5 | **66.3** |

*Table A6.* **Performance per task with pretraining and random initialization.** Results of MIL methods with random initialization (Base) and PC-108 supervised pretraining. The number of classes are specified below the task. The evaluation metrics for each task are indicated in parentheses. All models use UNI features as patch embeddings (Chen et al., 2024a). Performance on NSCLC subtyping is averaged across the internal TCGA cohort and the external NLST and CPTAC cohorts

| Task | Init. | ABMIL | CLAM | DFTD | DSMIL | ILRA | RRT | TransMIL | Transformer | WIKG | maxMIL | meanMIL |
|---|---|---|---|---|---|---|---|---|---|---|---|---|
| BRACS-C | Base | 64.1 (5.2) | 55.9 (5.4) | 64.3 (4.9) | 66.8 (4.9) | 67.0 (4.1) | 63.9 (3.2) | 64.2 (4.3) | 55.3 (3.7) | 60.1 (5.3) | 64.5 (4.0) | 61.6 (4.9) |
| $C=3$ | PC-108 | 68.5 (4.6) | 55.5 (5.6) | 68.4 (4.9) | 69.7 (4.9) | 67.3 (4.1) | 68.6 (4.4) | 62.8 (4.4) | 63.4 (5.0) | 77.4 (4.3) | 58.7 (4.5) | 52.8 (5.5) |
| BRACS-F | Base | 43.0 (3.9) | 31.2 (5.1) | 35.6 (4.5) | 40.2 (4.1) | 44.4 (3.8) | 43.5 (4.5) | 36.1 (3.3) | 18.3 (2.9) | 40.4 (4.2) | 34.2 (3.7) | 32.2 (4.2) |
| $C=7$ | PC-108 | 47.1 (5.3) | 33.7 (4.1) | 47.4 (4.5) | 44.2 (4.1) | 35.2 (3.8) | 36.2 (4.1) | 38.6 (5.6) | 42.9 (5.6) | 46.8 (4.7) | 39.0 (3.7) | 30.0 (4.1) |
| NSCLC | Base | 95.3 (0.6) | 91.1 (0.8) | 92.1 (0.7) | 94.0 (0.9) | 90.4 (0.7) | 93.9 (0.7) | 91.3 (0.8) | 93.3 (0.8) | 91.3 (0.8) | 95.9 (0.5) | 91.1 (0.7) |
| $C=2$ | PC-108 | 96.1 (0.6) | 92.0 (0.8) | 96.6 (0.7) | 95.3 (0.9) | 94.8 (0.7) | 94.7 (0.7) | 95.4 (0.7) | 94.4 (0.7) | 94.7 (0.7) | 95.4 (0.6) | 92.3 (0.7) |
| BCNB ER | Base | 85.7 (4.4) | 88.1 (3.0) | 85.3 (3.5) | 87.4 (2.7) | 87.4 (2.8) | 82.1 (4.0) | 87.4 (3.1) | 81.1 (3.7) | 90.1 (3.5) | 88.4 (3.4) | 86.3 (2.9) |
| $C=2$ | PC-108 | 88.6 (3.6) | 89.6 (2.7) | 93.3 (3.5) | 92.6 (2.7) | 92.8 (2.8) | 89.6 (2.9) | 90.6 (3.7) | 89.8 (3.7) | 89.0 (3.8) | 89.2 (3.0) | 89.5 (2.9) |
| BCNB HER2 | Base | 75.4 (4.8) | 72.4 (5.8) | 72.3 (6.0) | 72.5 (6.1) | 71.4 (5.5) | 71.9 (4.9) | 67.2 (5.2) | 65.8 (6.2) | 76.6 (4.7) | 73.9 (5.2) | 68.4 (6.0) |
| $C=2$ | PC-108 | 79.8 (4.2) | 74.4 (5.8) | 79.5 (6.0) | 74.0 (6.1) | 71.0 (5.5) | 72.6 (5.6) | 71.6 (5.9) | 61.9 (6.2) | 75.7 (5.8) | 70.3 (5.2) | 70.4 (6.0) |
| BCNB PR | Base | 81.5 (4.5) | 81.1 (5.5) | 78.3 (5.9) | 79.0 (4.8) | 78.1 (5.0) | 80.8 (5.0) | 76.1 (4.5) | 80.4 (4.4) | 83.6 (4.4) | 86.7 (4.8) | 79.9 (5.1) |
| $C=2$ | PC-108 | 85.2 (4.7) | 83.2 (5.2) | 86.4 (5.9) | 82.5 (4.8) | 80.3 (5.0) | 75.1 (5.4) | 85.7 (5.1) | 82.5 (5.3) | 89.6 (4.3) | 84.4 (5.1) | |
| BRCA ER | Base | 81.4 (3.5) | 80.7 (4.3) | 80.5 (2.8) | 82.9 (3.7) | 80.2 (3.6) | 81.8 (4.9) | 76.1 (4.1) | 76.0 (3.8) | 83.1 (3.9) | 83.6 (3.9) | 80.9 (4.4) |
| $C=2$ | PC-108 | 84.2 (2.8) | 81.8 (3.2) | 88.8 (2.8) | 83.5 (3.7) | 85.0 (3.6) | 86.3 (3.1) | 80.3 (3.5) | 84.2 (2.7) | 83.3 (3.8) | 84.8 (2.9) | 82.9 (3.5) |
| BRCA HER2 | Base | 63.2 (5.1) | 61.8 (5.0) | 63.0 (5.2) | 61.6 (5.8) | 64.4 (4.6) | 55.3 (6.0) | 55.8 (5.5) | 64.4 (5.0) | 53.5 (6.4) | 57.7 (5.5) | 51.7 (5.9) |
| $C=2$ | PC-108 | 68.6 (4.2) | 50.6 (5.7) | 54.1 (5.7) | 58.3 (5.8) | 71.2 (4.6) | 53.9 (5.1) | 60.0 (5.4) | 62.7 (5.0) | 58.1 (5.6) | 66.8 (5.5) | 62.6 (5.3) |
| BRCA PIK3CA | Base | 60.9 (4.0) | 59.6 (3.8) | 62.9 (3.9) | 62.7 (3.7) | 57.2 (3.5) | 57.7 (4.1) | 57.8 (4.5) | 44.6 (4.5) | 54.5 (4.2) | 55.2 (3.9) | 58.8 (4.4) |
| $C=2$ | PC-108 | 67.9 (4.0) | 60.3 (3.8) | 60.8 (3.9) | 63.0 (3.7) | 54.1 (3.5) | 60.0 (3.8) | 63.6 (4.5) | 63.9 (4.5) | 60.1 (4.4) | 59.9 (3.6) | 61.7 (4.3) |
| BRCA PR | Base | 71.4 (4.5) | 68.9 (4.6) | 71.0 (3.2) | 68.3 (2.9) | 72.9 (3.2) | 72.0 (4.2) | 69.2 (3.5) | 71.6 (3.8) | 73.0 (3.7) | 69.4 (5.0) | 73.5 (3.8) |
| $C=2$ | PC-108 | 77.6 (2.7) | 67.6 (3.5) | 79.1 (3.2) | 80.7 (2.9) | 78.5 (2.9) | 81.5 (3.7) | 78.4 (3.5) | 74.9 (3.3) | 74.8 (4.3) | 72.1 (3.2) | 70.9 (3.3) |
| EBRAINS-C | Base | 86.1 (2.1) | 86.1 (2.3) | 81.2 (1.9) | 86.2 (1.8) | 85.0 (1.8) | 86.7 (2.0) | 82.2 (2.1) | 86.6 (2.1) | 83.6 (2.1) | 82.0 (2.4) | 86.3 (2.3) |
| $C=12$ | PC-108 | 87.6 (1.8) | 87.2 (2.3) | 91.1 (1.9) | 88.1 (1.8) | 89.0 (1.8) | 88.5 (2.0) | 87.5 (2.4) | 90.7 (1.8) | 83.3 (2.1) | 89.0 (2.4) | 85.5 (2.3) |
| EBRAINS-F | Base | 67.1 (2.3) | 71.4 (2.0) | 61.5 (2.0) | 64.2 (2.1) | 71.3 (2.2) | 68.4 (1.8) | 62.1 (1.9) | 65.8 (2.1) | 66.6 (1.9) | 64.7 (2.1) | 69.3 (2.2) |
| $C=30$ | PC-108 | 69.2 (2.3) | 73.0 (1.9) | 69.4 (2.0) | 66.2 (2.1) | 64.6 (2.2) | 68.2 (1.9) | 67.6 (2.1) | 73.0 (1.8) | 70.1 (2.1) | 70.1 (2.2) | 72.3 (2.2) |
| GBMLGG-C | Base | 92.8 (1.8) | 91.7 (1.7) | 91.7 (2.2) | 93.1 (1.4) | 93.6 (1.6) | 92.2 (1.8) | 91.0 (1.2) | 92.9 (1.5) | 92.4 (1.2) | 94.3 (1.3) | 91.8 (1.6) |
| $C=2$ | PC-108 | 93.4 (1.2) | 92.8 (1.7) | 91.7 (2.2) | 95.7 (1.4) | 92.9 (1.6) | 93.0 (1.7) | 94.5 (1.1) | 91.5 (1.1) | 92.3 (1.4) | 93.4 (1.3) | 91.3 (1.3) |
| GBMLGG-F | Base | 44.9 (3.1) | 45.4 (3.8) | 43.9 (3.5) | 51.3 (3.2) | 53.6 (3.1) | 49.7 (3.3) | 48.3 (3.6) | 44.9 (3.2) | 41.6 (3.5) | 50.1 (3.4) | 47.2 (3.2) |
| $C=12$ | PC-108 | 53.7 (2.9) | 50.5 (3.8) | 57.1 (3.9) | 55.0 (3.2) | 47.5 (3.1) | 55.2 (3.5) | 49.2 (3.6) | 48.9 (3.9) | 43.4 (3.4) | 52.5 (3.4) | 47.0 (2.8) |
| NSCLC-EGFR | Base | 63.4 (7.0) | 60.6 (8.7) | 62.9 (9.5) | 64.8 (9.0) | 61.6 (8.8) | 62.3 (9.0) | 68.0 (7.0) | 63.3 (7.2) | 60.1 (8.5) | 64.0 (9.3) | 63.2 (8.1) |
| $C=2$ | PC-108 | 69.8 (7.0) | 62.6 (8.7) | 67.8 (9.5) | 60.1 (9.0) | 64.9 (8.8) | 69.1 (8.8) | 73.7 (7.0) | 69.9 (7.2) | 79.1 (6.1) | 57.8 (9.6) | 64.2 (8.1) |
| NSCLC-KRAS | Base | 60.0 (5.5) | 63.5 (5.4) | 58.1 (6.1) | 62.5 (7.4) | 52.9 (6.6) | 62.0 (5.7) | 59.3 (6.1) | 60.0 (6.5) | 54.0 (5.8) | 62.2 (6.3) | 63.6 (4.8) |
| $C=2$ | PC-108 | 65.2 (5.1) | 64.3 (5.4) | 71.4 (6.1) | 61.5 (7.4) | 60.8 (6.6) | 67.5 (6.2) | 66.6 (6.1) | 64.8 (4.5) | 62.3 (6.1) | 60.6 (6.3) | 57.3 (6.9) |
| NSCLC-STK11 | Base | 72.5 (3.6) | 69.9 (7.7) | 65.4 (7.2) | 71.5 (5.9) | 65.4 (6.6) | 72.4 (7.8) | 65.1 (7.1) | 68.8 (7.8) | 65.8 (6.9) | 70.8 (6.5) | 66.4 (6.4) |
| $C=2$ | PC-108 | 78.1 (3.6) | 72.3 (5.3) | 79.6 (7.2) | 71.5 (5.9) | 77.9 (6.6) | 82.7 (5.0) | 76.6 (7.1) | 73.6 (4.9) | 82.1 (7.1) | 61.8 (4.4) | 72.9 (5.9) |
| NSCLC-TP53 | Base | 73.7 (4.6) | 75.8 (4.4) | 69.5 (4.8) | 74.2 (5.0) | 71.4 (4.6) | 72.2 (5.1) | 72.1 (4.6) | 72.3 (5.4) | 69.7 (6.0) | 79.3 (4.7) | 72.2 (5.0) |
| $C=2$ | PC-108 | 77.9 (3.5) | 75.0 (4.6) | 78.3 (4.8) | 82.1 (5.0) | 80.2 (4.6) | 73.7 (5.5) | 80.5 (4.6) | 84.6 (3.4) | 76.3 (4.5) | 69.5 (4.7) | 80.2 (4.8) |
| PANDA | Base | 91.6 (0.7) | 91.2 (1.0) | 87.9 (0.7) | 91.2 (0.7) | 91.5 (0.9) | 91.1 (0.9) | 90.5 (0.9) | 90.4 (1.0) | 92.1 (0.8) | 89.7 (0.9) | 90.2 (1.0) |
| $C=7$ | PC-108 | 93.3 (0.5) | 91.8 (1.0) | 93.2 (0.7) | 93.5 (0.7) | 91.5 (0.9) | 91.5 (0.9) | 89.9 (0.8) | 90.7 (0.8) | 93.0 (0.8) | 88.9 (0.9) | 89.7 (1.0) |

