# OpenReview forum: "Do Multiple Instance Learning Models Transfer?"
_ICML.cc/2025/Conference — ICML 2025 spotlightposter_

### Official Review · Reviewer_5tA4 · 2025-03-06

**Overall Recommendation:** 4

**Summary:**

The paper presents the first comprehensive investigation into the transfer learning capabilities of MIL models in computational pathology. It evaluates 11 different MIL architectures pretrained on diverse pan‐cancer tasks (i.e., PC-108 and PC-43) across 19 downstream tasks, including cancer subtyping, grading, and biomarker prediction. The authors demonstrate that models initialized with supervised pan-cancer pre-training substantially outperform those with random initialization—even when there is a domain gap—and can even exceed the performance of state-of-the-art slide-level foundation models (e.g., CHIEF), all while using less pre-training data. The paper also explores the effect of model scale and few-shot learning performance, showing that simpler architectures (such as ABMIL) can be very competitive.

## update after rebuttal

After reviewing the authors' rebuttals, I appreciate their effort to compare with additional baselines and their insightful discussion on the beneficial transfer of features. Based on these improvements, I have raised my score to 4.

**Claims And Evidence:**

The paper makes several key claims:

(1) pre-training improves performance: Every evaluated MIL architecture shows a notable boost when initialized from a pan-cancer pretrained model compared to random initialization.

(2) Pan-cancer pre-training is robust: Models pretrained on pan-cancer datasets generalize well across different organs and task types.

(3) Model scaling matters: Larger models benefit more from effective pre-training and exhibit favorable scaling properties.

(4)Few-shot learning capability: Pretrained models perform exceptionally well in few-shot scenarios, indicating strong data efficiency.

These claims are backed by extensive experiments including comparisons via fine-tuning and KNN evaluations (e.g., in Table 1, Table 2, Figures 1–5), and through ablation studies that demonstrate the robustness.

**Essential References Not Discussed:**

One notable omission is the discussion of alternative supervised transfer baselines. For instance, incorporating insights from Raghu et al.’s “Transfusion: Understanding transfer learning for medical imaging” (NeurIPS 2019) could provide a more nuanced baseline than random initialization alone. Another baseline is simple mean-pooling to instance features. Addressing this could offer further insight into how supervised pre-training transfers to downstream tasks.

**Experimental Designs Or Analyses:**

The experimental design includes:

(1) Comprehensive Evaluations: Multiple MIL architectures are compared under the same conditions across a broad set of tasks.

(2) Ablation Studies: The paper investigates the effects of different pre-training tasks, model scales, and even different patch encoders.

(3) Few-shot Experiments: Repeated experiments with cross-validation ensure that the few-shot learning results are statistically meaningful.

One area for improvement is the inclusion of confidence intervals in key figures (e.g., Figure 3) and tables (e.g., Table 2) to assess the variability of the results.

**Methods And Evaluation Criteria:**

The authors adopt a rigorous experimental framework:

(1) pre-training and Transfer Protocol: MIL models are first trained on pan-cancer classification tasks and then evaluated on 19 downstream tasks using both end-to-end finetuning and frozen feature extraction (KNN).

(2) Evaluation Metrics: They employ AUROC for binary classification, weighted kappa for grading, and balanced accuracy for multiclass tasks.

(3) Diverse Datasets: The evaluation spans multiple orgons and task types, ensuring that the conclusions are not dataset-specific.

Overall, the methodology is appropriate and well-suited to address the problem of transfer in data-scarce clinical environments. However, adding statistical measures such as confidence intervals could improve the robustness of the evaluation.

**Other Comments Or Suggestions:**

(1) Table 2 Improvements: It would be valuable to include performance results for Prov-GigaPath along with confidence intervals to better capture performance variability.
(2) Figure 3 Enhancements: Adding confidence intervals in the few-shot performance plots would improve the statistical rigor of the presented results.

**Other Strengths And Weaknesses:**

Strengths:

(1) Practical Relevance: Demonstrating the effectiveness of pre-training with limited data is highly valuable for CPath.

Weaknesses:

Insight: The paper offers limited insight into why and how supervised pre-training leads to improved downstream performance.

Statistical Reporting: Key experimental results (e.g., Table 2, Figure 3) would benefit from the inclusion of confidence intervals.

Baseline Comparisons: Including comparisons with additional methods (e.g., “Transfusion”) could further contextualize the contribution.

**Questions For Authors:**

Can you provide more insight into which features or representations are most beneficial for MIL transfer? What do you hypothesize is being captured by the pan-cancer pre-training in MIL?

**Relation To Broader Scientific Literature:**

The paper is well-positioned within the existing literature on computational pathology and transfer learning. It draws on established MIL methods and compares its findings with those from slide-level foundation models and related transfer learning.

**Theoretical Claims:**

The paper is primarily empirical. There are no theoretical proofs provided.

---

> ### Author Rebuttal · Authors · 2025-04-01
>
> We thank the reviewer for their detailed feedback on areas to improve rigor and insights from our results.
>
> **Q1. Benchmarks and statistical measures**
> ***
> We have added benchmarks with finetuned GigaPath (Q1 of BdAq), as well as using CHIEF pre-training on the PC dataset (Q1 of BdAq). We show std due to character limits and will show confidence intervals in the final paper. Please see Q3 of Rh9m for std of few-shot experiments.
>
> **Q2. Baseline comparisons**
> ***
> We now include 1) mean pooling for all slide-level encoding experiments (GigaPath and CHIEF in BdAq Q1) and 2) selectively transferring early layers (motivated by [1]) Below, we report how different transfer methods affect finetuning performance on a pre-trained ABMIL model (three-layer MLP with gated attention aggregation). Following TransFusion's approach, we investigate finetuning performance as progressively fewer layers are transferred. Starting with the attention module, we re-initialize the attention layer to random weights (Reset Attn), the third linear layer + attention (Reset Lin3+), the second and third linear layer + attention (Reset Lin2+), and all weights (Reset All).
>
> We report performance relative to full weight transfer (PC108-L). Removing the attention layer causes the largest decrease (-5.0 average). Removing the remaining MLP layers results in a smaller but still substantial decrease (additional -3.3). These results highlight that the pretrained aggregation layer is crucial for successful supervised MIL pretraining. This contrasts with [1], which found that transfer of deeper layers has minimal effect on performance, emphasizing the unique nature of MIL transfer.
>
> |Task|PC108-L|Reset Attn|Reset Lin3+|Reset Lin2+|Reset All|Mean Pool|
> |-|-|-|-|-|-|-|
> | **Avg** |73.1| -5.0 | -5.2 | -6.6 | -8.3 | -12.5 |
> | BRACS-C |71.9| -8.5 | -8.5 | -12.8 | -11.0 | -16.2 |
> | BRACS-F |53.3| -12.1 | -10.5 | -10.4 | -10.1 | -23.1 |
> | GBM C |95.4| -1.2 | -1.0 | 0.0 | -0.2 | -3.3 |
> | GBM F |51.7| 0.0 | -0.8 | -1.8 | 0.8 | -2.4 |
> | Lung EGFR |76.1| -8.6 | -10.2 | -13.1 | -12.8 | -15.3 |
> | Lung KRAS |68.4| -2.0 | -4.5 | -5.8 | -8.7 | -8.1 |
> | Lung STK11 |86.7| 0.0 | -0.6 | -8.3 | -15.0 | -20.8 |
> | Lung TP53 |81.5| -7.3 | -5.9 | -0.5 | -10.0 | -10.8 |
>
> [1] Raghu, Maithra, et al. "Transfusion: Understanding transfer learning for medical imaging." Advances in neural information processing systems 32 (2019).
>
> **Q3. What features are most beneficial for transfer?**
> ***
> The experiment in Q2 suggests that the attention-based layer is the most beneficial for MIL transfer. To gain further insights, we next investigate the extent to which transferred layers change after finetuning. We follow the (SV)CCA approach used in [1], which measures the linear relationship between combinations of neuron activations across different models (0-100 scale). We compare activations before and after finetuning for each layer in standard (S) and large (L) ABMIL models on 30 randomly selected slides (45,232 patches) from the NSCLC-KRAS dataset.
>
> Results below show pretrained layers change substantially less over the course of training than randomly-initialized ones. Most notably, the third and final attention layer (abmil c), exhibits extremely low correlation (2.5 & 16.3 for ABMIL-L and -S) with its original layer activations after training with random initialization, while pretrained models maintain high similarity (88.7 & 96.6). Since abmil c is the critical attention layer responsible for converting each patch embedding into a scalar weight for slide-level aggregation, this finding suggests that MIL transfer benefits heavily from transferring learned aggregation strategies.
>
> |Layer|Base-L|PC108-L|Base-S|PC108-S|
> |-|-|-|-|-|
> |Lin 1|92.8 ± 18.3|95.5 ± 14.7|93.9 ± 16.0|97.2 ± 12.0|
> |Lin 2|81.2 ± 22.6|89.2 ± 17.9|||
> |Lin 3|47.2 ± 25.0|66.5 ± 26.3|||
> |abmil a|85.2 ± 24.7|88.6 ± 20.9|95.6 ± 13.6|92.9 ± 17.3|
> |abmil b|84.8 ± 25.1|88.7 ± 20.4|94.7 ± 14.7|96.6 ± 11.7|
> |abmil c|2.5 ± 0.0|82.7 ± 0.0|16.3 ± 0.0|97.7 ± 0.0|
> |Slide feat|37.7 ± 21.7|70.2 ± 32.9|77.8 ± 18.8|96.4 ± 4.1|
> |Average|61.6|83.1|75.7|96.2|
>
> **Q4. What do you hypothesize is being captured?**
> ***
> Based on our results indicating the transferability of the attention layer, we hypothesize that pan-cancer pre-training provides a better starting point for the MIL model, by focusing on cancerous regions and disregarding regions of low diagnostic importance. The characteristics of tumor regions, such as pleomorphic nuclei, increased cellular density, and entropic cellular arrangement are common motifs across various tasks. Meanwhile, regions such as smooth muscle, processing artifacts, and red blood cells are consistently of minimal diagnostic relevance. Whereas training an MIL model from scratch leads to challenges learning to disregard these background patches, our results suggest that models trained on a large pan-cancer classification task are already equipped to prioritize tumor morphologies prior to finetuning.

---

### Official Review · Reviewer_BdAq · 2025-03-07

**Overall Recommendation:** 3

**Summary:**

This paper investigates transfer learning in MIL models for computational pathology. The authors test 11 MIL models across 19 pretraining tasks, showing that finetuning pretrained models significantly outperforms training from scratch, despite domain differences. Pan-cancer pretraining enables consistent generalization across organs and tasks, surpassing SOTA models. The findings highlight MIL models' adaptability and the advantages of pretraining in computational pathology.

## update after rebuttal
After careful consideration of all comments and the corresponding responses, I'll retain the score.

**Claims And Evidence:**

The paper’s claims are well-supported by clear and convincing evidence. The authors thoroughly evaluate 11 MIL architectures across 19 tasks. The results are presented in figures and tables demonstrate significant performance gains from pretraining.

**Essential References Not Discussed:**

The paper provides a comprehensive overview of related work in MIL and slide foundation models.  However, the inclusion of foundational models, such as Virchow, could be beneficial.

**Experimental Designs Or Analyses:**

The experimental designs and analyses are sound.

**Methods And Evaluation Criteria:**

The proposed methods and evaluation criteria are appropriate for the research question.

**Other Comments Or Suggestions:**

The manuscript contains several typographical errors, including misaligned citations in the literature review on page 2, lines 077-089, and inaccuracies in the publication years of classic Transformer references.

**Other Strengths And Weaknesses:**

Strengths:
- The paper is well-written and clearly structured.
- The experimental methodology is rigorous and comprehensive.
- The authors provide a thorough analysis of the results and relate them to the broader literature.

Weaknesses:
- The paper primarily emphasizes empirical findings, with limited contributions to technical or methodological advancements.
- The paper's extensive experimental results corroborate the well-established principle that initialization with pre-trained parameters outperforms random initialization.

**Questions For Authors:**

Surprisingly, the paper's methodology outperformed the SOTA CHIEF. The authors attribute this, in part, to their meticulously curated pretraining dataset. It would be intriguing to observe the results of training CHIEF using the authors' dataset.

**Relation To Broader Scientific Literature:**

It highlights the contrast between the extensive research on MIL architecture development and the lack of investigation into MIL model transfer in computational pathology. The authors also discuss the relationship between their work and the development of slide foundation models, positioning supervised MIL transfer as a simple and effective alternative.

**Theoretical Claims:**

No theoretical claims involved.

---

> ### Author Rebuttal · Authors · 2025-04-01
>
> We thank the reviewer for their valuable suggestions to further explore slide foundation models and clarify contributions. We provide our response below.
>
> **Q1. Retraining CHIEF**
> ***
> We train a new model (PC-CHIEF) on the PC dataset using CHIEF's training recipe, comprised of supervised contrastive loss and CLIP-based text embeddings of anatomical site.
>
> Our evaluation shows that PC-CHIEF achieves a mean performance of 69.2 across tasks, which is slightly lower than the original CHIEF's 69.8. This similar performance suggests that dataset quality is not the primary factor explaining the performance gap between the PC-108 model and CHIEF. Notably, PC-CHIEF underperforms our PC-108 model by a large margin, despite being trained on the same data samples with additional textual embeddings and contrastive loss. This result highlights our pre-training approach as a simple but effective means of developing highly transferrable MIL models. We show performance averaged across tasks within each dataset, with number of tasks indicated in parentheses. All results use CTransPath features.
>
> |Task|PC-108|CHIEF|PC-CHIEF|Base|Mean Pool|
> |-|-|-|-|-|-|
> |Avg (14)| 70.8|68.8|68.6|68.1|61.3|
> | BRACS (2)| 60.3 ± 4.5 | 58.3 ± 5.0 | 57.2 ± 5.0 | 54.4 ± 4.9 | 38.1 ± 4.3 |
> | BRCA (4)| 74.1 ± 4.0 | 71.4 ± 4.2 | 72.1 ± 3.9 | 71.5 ± 4.8 | 69.4 ± 4.7 |
> | NSCLC (4)| 71.8 ± 7.2 | 68.4 ± 6.4 | 67.9 ± 6.3 | 68.1 ± 6.9 | 62.5 ± 6.4 |
> | EBRAINS (2)| 68.7 ± 2.2 | 70.9 ± 2.2 | 70.1 ± 2.4 | 68.9 ± 2.1 | 57.2 ± 2.2 |
> | GBMLGG (2)| 74.6 ± 2.5 | 72.8 ± 2.6 | 72.8 ± 2.5 | 73.9 ± 2.5 | 69.6 ± 2.6 |
>
> To further explore the efficacy of supervised pretraining, we extended our investigation to a different slide FM, Gigapath [1]. Unlike CHIEF, Gigapath was trained using a fully self-supervised approach, using a dataset of 171,189 WSIs. We compare finetuning performance of Gigapath with ABMIL pretrained on PC-108. All results use Gigapath patch features.
>
> |Dataset|PC-108|Gigapath|Base|Mean Pool|
> |-|-|-|-|-|
> |Average (14)|73.0|71.9|71.5|67.0|
> |BRACS (2)|58.7 ± 4.7|54.6 ± 4.3|59.3 ± 4.6|40.6 ± 4.7|
> |EBRAINS (2)|77.6 ± 2.1|79.3 ± 2.1|77.7 ± 2.0|80.9 ± 1.7|
> |GBMLGG (2)|70.9 ± 2.4|73.9 ± 2.4|70.0 ± 2.4|69.6 ± 2.6|
> |NSCLC (4)|76.4 ± 5.5|71.7 ± 6.4|70.3 ± 4.9|66.0 ± 6.2|
> |BRCA (4)|73.0 ± 4.3|73.0 ± 4.3|72.8 ± 4.4|70.7 ± 4.4|
>
> The ABMIL pretrained model, with a pretraining set only 2% the size of Gigapath, demonstrated substantially higher performance compared to the Gigapath slide FM. These results underscore the effectiveness of our pretraining approach even when trained with a substantially less data.
>
> [1] Xu, Hanwen, et al. "A whole-slide foundation model for digital pathology from real-world data." Nature 630.8015 (2024): 181-188.
>
>
>
> **Q2. Limited contributions to methodological advancements.**
> ***
> While our work does not propose technical novelty in the form of a new MIL architecture, we respectfully disagree that our work makes limited contributions to methodological advancement. We note that in general ML, most high-impact papers on transfer learning are published using hypothesis-driven experimentation that emphasize unique scientific insights instead of methodology as a research contribution (Line 077-089). Not only is empirical investigation on transfer learning a valid and important research direction, but we also emphasize that research on MIL transfer is almost entirely absent in CPath. The vast majority of CPath studies in ML/CV conferences train MIL models from scratch, which may stem from lack of understanding on when and where these models would benefit from transfer. Furthermore, we provide an accessible, supervised alternative to slide foundation models trained on massive WSI cohorts (60-200k) via self-supervised learning, which demand substantial data and computing resources.
>
>
> **Q3. Limited Novel Insights**
> ***
> Though the benefits of transfer learning are well-established in the broader ML community, the absence of existing works on MIL transfer signals a distinct gap in the literature. As the first work to investigate MIL transfer, we reveal pan-cancer pretraining (considered by qGZD as a novel technique) as a powerful and overlooked approach to improve performance across all tasks and MIL methods, allowing a pre-trained ABML model to outperform SOTA slide foundation models (CHIEF and GigaPath) while requiring only 2-7% of the training samples. In addition, we also provide practical insights specific to MIL transfer, such as the importance of different layers for transfer (see response to 5tA4 Q2-3), the effect of model size on transfer (Rh9m Q1), the effect of task difficulty (Rh9m Q4), and the transferability of models across different organs (Figure A1).
>
> **Q4. Textual edits**
> ***
> We agree that inclusion of patch foundation models will provide a comprehensive overview of related works. We will include mention of models such as Virchow in the revision. We will adjust the typesetting and reference issues in the revision.

---

> > ### Comment · Reviewer_BdAq · 2025-04-07
> >
> > Thank you for the responses, which have largely addressed my concerns. After careful consideration of all comments and the corresponding responses, I'll retain the score.

---

> > > ### Author Response · Authors · 2025-04-08
> > >
> > > We thank the reviewer for their feedback and are pleased to have addressed their concerns.

---

### Official Review · Reviewer_qGZD · 2025-03-11

**Overall Recommendation:** 3

**Summary:**

The paper investigates the transfer learning capabilities of Multiple Instance Learning (MIL) models in computational pathology, evaluating 11 MIL models across 19 tasks. It finds that pretrained MIL models consistently outperform those initialized randomly, with pan-cancer pretraining tasks (such as PC-108 and PC-43) showing substantial performance gains across various downstream tasks. This pan-cancer pretraining enhances generalization across different organs and task types, often outperforming single-disease pretraining. Simple MIL architectures, like ABMIL, demonstrate high transfer performance, sometimes surpassing more complex transformer-based models. Larger models, particularly those based on transformers, benefit more from pretraining, showing significant performance improvements. The study also highlights the few-shot learning capabilities of pan-cancer pretrained models, which perform well even with limited data. Additionally, the benefits of pan-cancer pretraining are consistent across different patch encoders, indicating the robustness of the MIL framework. The findings suggest that supervised MIL models exhibit strong adaptability and transferability, with pan-cancer pretraining emerging as a highly effective strategy for enhancing performance in computational pathology.

## update after rebuttal
Thanks for the authors, I am raising my score to 3.

**Claims And Evidence:**

The claims in the submission are well-supported by empirical evidence. The authors demonstrate that MIL models pretrained on diverse pan-cancer tasks consistently outperform those initialized randomly across various downstream tasks, highlighting the benefits of pan-cancer pretraining for generalization and few-shot learning. Simple architectures like ABMIL show high transfer performance, while larger models, particularly transformers, benefit more from pretraining. The findings are robust across different patch encoders, indicating the MIL framework's inherent transferability. However, further investigation into comparisons with other pretraining strategies, the impact of dataset quality and diversity, and the generalizability to other domains could provide additional insights.

**Essential References Not Discussed:**

In my opinion, "How well do self-supervised models transfer?" is worth citing.

**Experimental Designs Or Analyses:**

The experimental design and analysis in the paper are sound and well-structured. The authors comprehensively evaluate 11 MIL models across 19 diverse tasks and datasets, providing robust empirical evidence for the transfer learning capabilities of MIL models. The use of pan-cancer pretraining tasks (PC-43 and PC-108) is a novel approach that effectively demonstrates enhanced generalization across different organs and task types. The dual evaluation settings (end-to-end finetuning and KNN) and the inclusion of few-shot learning experiments further strengthen the findings.

### Potential Issues

Comparison with Other Pretraining Strategies: The study focuses on supervised pretraining but lacks comparisons with self-supervised learning methods, which could provide a more comprehensive understanding.
Statistical Significance: Detailed statistical significance tests for performance differences are missing, which could further validate the robustness of the results.
Overall, the experimental design is robust, but addressing these potential issues could enhance the study's comprehensiveness and practical applicability.

**Methods And Evaluation Criteria:**

The proposed methods and evaluation criteria are highly suitable for the problem of assessing MIL model transferability in computational pathology. The comprehensive evaluation of multiple MIL models across diverse datasets and tasks effectively addresses the research questions and provides robust insights. The use of pan-cancer pretraining and standardized metrics ensures that the findings are generalizable and meaningful. While the methods are well-designed, further inclusion of self-supervised learning comparisons and clinical validation could enhance the study's comprehensiveness and practical relevance. Overall, the approach is well-aligned with the goals of the research.

**Other Comments Or Suggestions:**

Innovation in Pretraining: Consider exploring more innovative pretraining strategies beyond pan-cancer datasets. For example, integrating multi-modal data (e.g., combining histopathology images with clinical data) could offer new insights.

Novel Architectures: Investigate novel MIL architectures or modifications to existing ones that could enhance transferability. This could include exploring attention mechanisms or graph-based models.

Cross-Domain Transfer: Explore cross-domain transfer learning, such as transferring knowledge from natural images to histopathology images. This could provide a more comprehensive understanding of transfer learning in computational pathology.

Benchmarking: Develop new benchmarks or datasets that better reflect real-world clinical scenarios. This could help in evaluating the robustness and generalizability of the proposed methods.

**Other Strengths And Weaknesses:**

### Strengths

1. Comprehensive Evaluation: The paper thoroughly evaluates 11 MIL models across 19 diverse tasks, providing robust and generalizable findings.

2. Pan-Cancer Pretraining: Introducing pan-cancer pretraining tasks (PC-43 and PC-108) is a novel approach that significantly enhances model generalization.

3. Few-Shot Learning: The study explores few-shot learning scenarios, demonstrating the practical applicability of pan-cancer pretraining in data-scarce environments.

4. Model Scalability: The analysis of model size and transfer performance offers valuable insights into the scalability of MIL models.
Practical Implications: The findings have significant practical implications for improving performance in computational pathology with limited data.

### Weakness:

1. Limited originality: The main drawback of this paper is the lack of substantial originality. While the application of pan-cancer pretraining is novel, the overall approach largely builds on existing concepts in transfer learning and MIL.

2. Single visualization scheme: the author mainly uses heat map to show the effect, but does not include the visualization results of specific tasks. An example is the effect comparison of segmentation tasks.

**Questions For Authors:**

No more questions here.

**Relation To Broader Scientific Literature:**

The key contributions of the paper are closely related to the broader scientific literature on transfer learning and pretraining strategies in computational pathology. Specifically:

1.Transfer Learning in Computational Pathology: The paper investigates the transfer learning capabilities of MIL models, extending the well-established concept of transfer learning from other domains like natural language processing and computer vision to computational pathology.

2.Pan-Cancer Pretraining: The use of pan-cancer pretraining tasks to enhance model performance aligns with the trend of leveraging large, diverse datasets for pretraining, similar to strategies used in models like BERT and GPT. This approach is shown to improve generalization and transferability in MIL models.

3.Model Scalability: The finding that larger models, particularly transformers, benefit more from pretraining is consistent with the broader literature on model scalability. This highlights the importance of effective initializations for complex models.

4.Few-Shot Learning: The paper's exploration of few-shot learning scenarios is relevant to addressing data scarcity in computational pathology, a common challenge in the field. The results demonstrate the potential of transfer learning to improve performance with limited data.

Overall, the paper builds on existing knowledge in transfer learning and pretraining, providing specific insights into the application of these strategies in computational pathology using MIL models.

**Theoretical Claims:**

The paper does not include any theoretical proofs, and the claims are based on extensive empirical evaluations. The empirical results are robust and provide strong support for the claims made. While theoretical analysis could further strengthen the findings, the current approach is well-aligned with the goals of the study and provides valuable insights into the transfer learning capabilities of MIL models in computational pathology.

---

> ### Author Rebuttal · Authors · 2025-03-31
>
> Thank you for your detailed feedback and for sharing your enthusiasm on the robust empirical evidence and strong performance of transferring MIL models. We have sought to address nearly all suggestions in additional experiments. Further details are provided below.
>
> **Q1. Pretraining strategies**
> ***
> To address the reviewer’s request, in addition to comparing against CHIEF, a SOTA slide foundation model (FM) trained on 60k WSIs (Table 2), we also:
> 1) Compared against Gigapath, another SOTA slide FM trained on 171k WSIs. We show similar findings that ABMIL pretrained on PC-108 (3,499 WSIs) outperforms Gigapath (see BdAq Q1).
>
> 2) Implemented CHIEF’s vision-language (VL) pretraining on PC-108 for fair comparison on pretraining strategy. Our results indicate conventional supervised pretraining outperforms VL pretraining on the same dataset (see BdAq Q1).
>
>
> **Q2. Limited originality**
> ***
> We respectfully disagree regarding the lack of substantial originality. Despite significant interest in MIL architectures for computational pathology (CPath) in ML/CV conferences, there is little to no investigation on MIL transfer in CPath. This is in stark contrast to progress made in general ML, with many high-impact papers using hypothesis-driven experimentation that contribute unique scientific insights (Line 077-089) without introducing new architectures [1,2]. Despite the importance of transfer learning in ML broadly, pretrained MIL model transfer remains unexplored in CPath due to limited understanding of its success conditions (Line 057-076, 102-109).
>
> As the first work to investigate supervised MIL transferability in CPath, **we believe that this research question is not only substantially original, but also impactful by highlighting a new avenue for obtaining highly generalizable slide-level representations**. Despite progress made with self-supervised slide FMs, our alternative approach of supervised MIL model transfer has not been previously explored (Line 090-101). We show that our approach outperforms these methods while using less than 10% of the training data. Furthermore, we provide extensive insights into MIL transfer, such as how which features are transferred, how important each layer is to transfer, and how model size, patch encoders, MIL architecture, and task difficulty affect transfer.
>
> [1] Kornblith, Simon, et al. "Do better imagenet models transfer better?" CVPR (2019): 2661-2671
>
> [2] Fang, Alex et al. "Does progress on ImageNet transfer to real-world datasets? Neurips 36 (2024)
>
> [3] Ericsson et al. "How well Do Self-Supervised Models Transfer?" CVPR (2021): 5414-5423
>
> **Q3. Visualization**
> ***
> We agree further interpretability is valuable. To this end, we also provided t-SNE visualizations showing pretrained slide embeddings differentiate classes better than random initialization (Figure 5). Since the scope of this work is on investigating MIL models, we preclude segmentation tasks, which focus on categorizing pixels rather than slides.
>
> Motivated by the reviewer’s suggestion, we generated additional attention-based heatmaps for ABMIL on BRACS and NSCLC subtyping, finding that pretrained models focus on diagnostically relevant tumor regions even before finetuning. This indicates that the aggregation layer is highly transferable between tasks, which we further validated through quantitative explainability experiments (see 5tA4 Q3-4). Updated visuals and discussions will be included in our final submission.
>
> **Q4. Novel architectures**
> ***
> We have conducted extensive experimentation on 11 MIL architectures, with one of them also being graph-based (WikG). Though novel architectures are not the focus of our study, we modified a few MIL architectures to further explore the impact of model size on transferability (see Rh9m Q1). Results showed increased performance with larger model sizes compared to training from scratch, encouraging exploration of new high-capacity MIL models despite limited dataset size.
>
> **Q5. Cross-domain transfer**
> ***
> While investigating transfer from natural to pathology images is interesting, there is a scarcity of giga-pixel natural images to properly assess MIL model transfer. Instead, we had included transfer results using ResNet50 patch feature encoder (Table 3) trained on natural images, which showed significantly poorer performance than in-domain patch feature encoders.
>
> **Q6. Other comments**
> ***
> *Novel Datasets*: As our work evaluates 11 MIL models across 21 pre training tasks, contributions such as novel benchmark datasets could not be fully explored, though we also note that our study is potentially one of the largest in ML/CV conferences. *Statistical tests*: We provide standard deviation via 100 trials of bootstrapping for our updated results (see Bda4 Q1, 5tA4 Q3, Rh9m Q1, Q3) and will provide confidence intervals in the main text. *Suggested reference*: We clarify that "How well do self-supervised models transfer?" was cited in our submission (Line 082-083).

---

> > ### Comment · Reviewer_qGZD · 2025-04-04
> >
> > Thanks for the authors, I am raising my score to 3.

---

> > > ### Author Response · Authors · 2025-04-06
> > >
> > > We are happy to hear we addressed your concerns and sincerely thank you for your thorough and constructive review of our submission.

---

### Official Review · Reviewer_RH9m · 2025-03-13

**Overall Recommendation:** 4

**Summary:**

This work explores the transferability of multiple instance learning in computational pathology. A variety of experiments are conducted to investigate how various factors affect the transferability, filling the gap of CPath community.

**Claims And Evidence:**

Some experimental results need further justification.

1. Given that highly parameterized models, e.g. transformer-based models, benefit more from pretraining, why does ABMIL perform best?

2. Although the OT-108/43 task for pretraining demonstrates good transferability in KNN-based adaptation for target tasks (indeed, this does not hold in the KNN performance of ABMIL, see Figure A2), there seems to be no similar conclusions in finetuning performance (Figure A1). As such, I am concerned about how reliable the conclusions drawn from the results are in Figure 2. Please justify it.

**Essential References Not Discussed:**

Regarding different patch encoders, the previous work mSTAR [1] had a similar experiment, where they claimed that the pretrained aggregator paired with the poor patch extractor benefits more from pretraining. I wonder if pretraining MIL still works when the patch extractor is strong enough, which can be validated in the patch encoder of Virchow 2G, a SOTA strong patch encoder.

[1] A Multimodal Knowledge-enhanced Whole-slide Pathology Foundation Model, arxiv, 2024.

**Experimental Designs Or Analyses:**

1. Data contamination. Among 19 tasks, several tasks have the same data although labels are different. In this case, I am not sure if the test sets of target tasks are used for pretraining. For example, has the test set of NSCLC-KRAS been included in the pretrain data of NSCLC-TP53? If so, we cannot tell which factor contributes to the performance gain, pretraining itself or data contamination.

2. The details of few-shot experiments should be presented. What target tasks are in Figure 3? I am concerned that if the model has already seen the samples of target classes during pretraining, which should not be viewed as few-shot learners.

3. The authors claimed that “This diverse and challenging pretraining task likely promotes the learning of comparatively more detailed, generalizable slide-level representations.” Given that an individual dataset (e.g. NSCLC) has multiple tasks, have you investigated if the model can benefit more from multi-task pretraining, which is supposed to be a more challenging pretraining task?

4. Given that the transferability is strongly related to the difficulty of pretrain tasks, I recommend authors to separate pretrain tasks into two groups, easy and difficult, and see if there are significant differences in the performance of the two groups.

5. The authors clarified that highly parameterized architectures benefit more from pretraining. Different parameter sizes of the transformer should be investigated. I wonder if the reason why the transformer underperforms ABMIL is due to insufficient parameters.

6. From Figure 4, we can see that performance continues to increase monotonically with the increase in parameters. In this case, to fully explore the scalability of model size, why not continue to increase the parameters until no further improvement is observed?

**Methods And Evaluation Criteria:**

This research fills the gap in the field of CPath and provides valuable guidance for developing MIL approaches in the future.

**Other Comments Or Suggestions:**

- There is a mistake in the order of BRACS-C and BRACS-F in Figure 2.
- The availability of code and data PC-108/43 can contribute to the reproduction of the conclusion in this work.

**Other Strengths And Weaknesses:**

## Strength
1. Comprehensive experiments are designed for the investigation of various factors contributing to the transferability of MIL for CPath.
2. This research fills the gap in the field of CPath and provides a valuable guidance for developing MIL approaches in the future.
3. The paper is well-written and easy to follow.

## Weakness
please see 'Claims And Evidence' and 'Experimental Designs Or Analyses'.

**Questions For Authors:**

please see 'Claims And Evidence' and 'Experimental Designs Or Analyses'.

**Relation To Broader Scientific Literature:**

Despite the significant research interest in the development of MIL architectures and the well-known advantages of transfer learning in general machine learning, there has been almost no investigation into the effectiveness of MIL models in transferring knowledge in CPath. This work fills this gap and provides valuable guidance for developing MIL approaches in the future.

**Theoretical Claims:**

No theoretical proofs in this work.

---

> ### Author Rebuttal · Authors · 2025-04-01
>
> We thank the reviewer for their extensive suggestions on further improving the rigor and insights from our work.
>
> **Q1. Model Size**
> ***
> We add larger ABMIL models (7M and 9M parameters) and Transformers at comparable sizes, finding:
> 1) ABMIL performance plateaus in the 5-7M range, with a notable decrease at 9M parameters.
> 2) At smaller sizes (0.2M and 1M), ABMIL and Transformer achieve similar performance both with and without pretraining.
> 3) At larger sizes, Transformers benefit substantially from pretraining, even outperforming ABMIL at 7M. Similar to ABMIL, Transformers show reduced performance at 9M.
>
> We display the performance averaged across BRCA ER/PR, NSCLC TP53/STK11, GBM C/F, BRACS C/F.
> |Approx Params|ABMIL|Transformer|ABMIL PC108|Transformer PC108|
> |-|-|-|-|-|
> |0.2M|70.5|67.4|73.1|72.9|
> |1M|70.1|68.8|72.6|72.6|
> |2.6M|-|64.4|-|70.5|
> |3.2M|71.3|65.2|74.0|71.2|
> |5.2M|70.7|65.1|75.7|73.8|
> |7M|71.3|66.3|75.4|76.8|
> |9M|70.6|67.5|74.1|72.1|
>
> **Q2. Why does ABMIL perform best?**
> ***
> ABMIL is consistently effective across CPath tasks and patch encoders [1]. In the data-restricted regimes common in CPath, the default transformer configuration may be overparameterized. Our results show that transformers can achieve comparable performance to ABMIL at smaller parameter counts and through pretraining at larger model sizes.
>
> [1] Campanella, Gabriele, et al. "A clinical benchmark of public self-supervised pathology foundation models." arXiv (2024).
>
> **Q3. Few-shot**
> ***
> The target tasks in Figure 3 included molecular (NSCLC TP53/STK11/EGFR, BCNB ER/PR/HER2, GBMLGG C) and morphological classification (BRACS C/F). We included BRACS C/F, as PC-108 contains only invasive carcinoma cases (a single label in BRACS F). We recognize this could raise concerns and will exclude BRACS in the revision.
>
> Shown below is performance averaged over ABMIL, TransMIL, Transformer, DFTD, and CLAM for molecular tasks alone and BRACS alone, confirming that few-shot learning benefits are robust to tasks without any label overlap.
>
> |k|MOL-PC108|MOL-PC43|MOL-base|BRACS-PC108|BRACS-PC43|BRACS-base|
> |-|-|-|-|-|-|-|
> |4|57.1 ± 5.2|52.8 ± 4.6|52.7 ± 2.9|35.4 ± 5.7|35.6 ± 4.4|26.8 ± 5.1|
> |16|64.1 ± 4.9|60.2 ± 4.8|56.4 ± 3.8|45.8 ± 4.5|44.2 ± 4.3|36.2 ± 4.3|
> |32|70.1 ± 4.2|66.9 ± 5.0|61.7 ± 4.4|45.9 ± 4.0|46.8 ± 4.2|39.1 ± 5.4|
>
> **Q4. Task Difficulty**
> ***
> To compare transfer quality between easy and hard tasks, we perform a paired t-test on transfer performance between the easiest and hardest task for each dataset. Given a fixed pretraining dataset, this allows us to investigate how the difficulty of the training objective affects the transferability of the final model. For morphological datasets, we assign coarse and fine subtyping as easy and hard tasks. For molecular datasets, which typically have multiple tasks (e.g BCNB ER/PR/HER2), we compare the model performance (averaged over ABMIL, DFTD, TransMIL, Transformer trained from scratch) on each task, selecting the task with the lowest and highest AUC as the hard and easy task, respectively. This resulted in the following:
> ||BRACS|EBRAINS|GBM|PC|BCNB|BRCA|NSCLC|
> |-|-|-|-|-|-|-|-|
> |Easy|C|C|C|43|ER|ER|TP53|
> |Hard|F|F|F|108|HER2|PIK3CA|KRAS|
>
> For each pair of pretraining tasks, we compare finetuning performance across the 17 remaining evaluation tasks for four MIL models, resulting in a total of 476 paired points (4 models x 7 pretraining x 17 evaluation).
> The result shows that **pretraining on hard tasks leads to better transfer performance, with average improvement of +0.5 (95% CI=0.1-0.9, p=0.017)**. This indicates that challenging training objectives enhance the transferability of the final model.
>
> **Q5. PC Transferability**
> ***
> Although ABMIL shows more variability in KNN performance, Figure A2 shows that PC pretraining consistently achieves strong average performance across all models. Specifically, PC-108 ranks second-highest in ABMIL and highest in both TransMIL and Transformer KNN evaluations. Figures 2 and A2 both therefore support our conclusion that PC pretraining produces robustly transferable features.
>
> Our finetuning results in Figure A1 further validate this finding, showing that although there is some variability, PC pretraining consistently outperforms random initialization and achieves the highest average performance among pretraining tasks across all models, including ABMIL. Thus, we believe our conclusion from Figure 2 is well validated by our finetuning results. We will make this connection clearer in the final submission.
>
> **Q6. Suggestions**
> ***
> *Multitask*: We agree that multi-task pretraining is a promising next step and will investigate this in further iterations. *Data Splits*: We share splits between tasks from the same dataset to ensure fair comparisons. *Patch encoders*: We will add Virchow 2G results in the final paper. *Data availability*: To facilitate further work on this topic, we will release the model weights and code for PC108/43 pretrained models.

---

> > ### Comment · Reviewer_RH9m · 2025-04-04
> >
> > Thanks to the authors' efforts, which addressed most of the concerns. Due to the inconsistency in KNN results and the insufficient exploration of patch encoders, I decide to maintain my score. The former weakens the credibility of the conclusions, while the latter may lead to situations where these patterns found in this work no longer hold when patch encoders are sufficiently strong.

---

> > > ### Author Response · Authors · 2025-04-06
> > >
> > > Thank you for your valuable feedback and for recognizing our efforts in addressing the concerns. We appreciate your thoughtful evaluation.

---

### Decision · Program_Chairs · 2025-05-01

**Decision:**

Accept (spotlight poster)

**Comment:**

This paper examines whether fine tuning MIL digital pathology models using transfer learning helps in other downstream tasks and provides an affirmative answer. Although the finding is not surprising, the breadth of the experiments and the number of methods tested are impressive. All reviewers recommend acceptance and the AC concurs